# Multilingual Coarse Political Stance Classification of Media. The Editorial Line of a ChatGPT and Bard Newspaper

**Cristina España-Bonet**
DFKI GmbH, Saarland Informatics Campus
Saarbrüken, Germany
`cristinae@dfki.de`

## Abstract

Neutrality is difficult to achieve and, in politics, subjective. Traditional media typically adopt an editorial line that can be by their potential readers as an indicator of the media bias. Several platforms currently rate news outlets according to their political bias. The editorial line and the ratings help readers in gathering a balanced view of news. But in the advent of instruction-following language models, tasks such as writing a newspaper article can be delegated to computers. Without imposing a biased persona, where would an AI-based news outlet lie within the bias ratings? In this work, we use the ratings of authentic news outlets to create a multilingual corpus of news with coarse stance annotations (Left and Right) along with automatically extracted topic annotations. We show that classifiers trained on this data are able to identify the editorial line of most unseen newspapers in English, German, Spanish and Catalan. We then apply the classifiers to 101 newspaper-like articles written by ChatGPT and Bard in the 4 languages at different time periods. We observe that, similarly to traditional newspapers, ChatGPT editorial line evolves with time and, being a data-driven system, the stance of the generated articles differs among languages.

## 1 Introduction

Instruction-following language models (ILMs) are omnipresent. Their use is not so extended as that of search engines yet, but due to the availability and high quality of systems and models such as Alpaca (Taori et al., 2023), Bard (Google, 2023), BLOOMZ and mT0 (Muennighoff et al., 2023), ChatGPT (OpenAI, 2023), Llama 2-chat (Touvron et al., 2023), or Koala (Geng et al., 2023), their use is expected to be more common in the near future.

These models face several problems being the most relevant the lack of trustworthiness (van Dis et al., 2023; Huang et al., 2023; Wang et al., 2023a). They are not ready to be used as a source of reliable information if their outputs are not fact checked. A second big issue with systems based on language models (LM) is the fact that they might reproduce the biases present in the training data (Navigli et al., 2023). Biases range from cultural missrepresentation due to data imbalance to offensive behaviour reproduced from written texts. LMs are finetuned into ILMs either in a supervised way using input-output pairs and an instruction (Wei et al., 2022; Wang et al., 2022, 2023b) or with reinforcement learning from human feedback (Ouyang et al., 2022; Nakano et al., 2021). In both cases, the finetuning should help removing bias. But neutrality is something very difficult to achieve, also for the humans that generate the supervisory data. The finetuning phase might therefore *over correct* the original biases or introduce new ones. For methods that generate the supervision data with the LM itself, the original biases might be inherited.

We focus on a specific use of ILMs: the writing of newspaper articles. Journals and newspapers follow an editorial line which is in general know to the reader. Besides, sites such as AllSides,[1] Media Bias Fact Check[2] (MB/FC), or Ad Fontes Media[3] provide ratings about the political bias of (mostly USA) media sources and their quality with respect to factual information. With these ratings, conscientious readers can make informed decisions about which media outlets to choose in order to get a balanced perspective. But what happens when journalists use systems such as ChatGPT or Bard to aid in their writing? As said above, humans also have biases, the danger lies in being unaware of them, as they might affect the user's/reader's perspective (Jakesch et al., 2023; Carroll et al., 2023). ChatGPT already warns its users about misinformation. However, the political bias, if any, is not known apart from the subjective perception that a user has.

---

[1] `https://www.allsides.com/`
[2] `https://mediabiasfactcheck.com/`
[3] `https://adfontesmedia.com/`

We address the question above for articles generated by ChatGPT and Bard in four languages: English, German, Spanish and Catalan. We do this in an automatic and systematic way with almost no human intervention so that the method can be easily extended to new languages and other ILMs with few effort. We do not aim at classifying individual articles with their specific bias, but to classify the media source (an ILM in this case) as Left or Right-oriented in a similar way as the media bias sites do for newspapers and other media outlets.

## 2 Corpora Compilation

We approach our task as a classification problem with two classes: Left ($\mathbb{L}$) and Right ($\mathbb{R}$) political orientations. This is a simplification of the real problem, where articles can also be neutral and there might be different degrees of biases. Previous work relied on 3 or 5 classes, always including the neutral option (Baly et al., 2020; Aksenov et al., 2021). In these works, data was manually annotated creating high quality training data but also limiting a lot the scope of the work in terms of languages and countries covered. When using the fine-grained classification scale, the authors acknowledge a bad generalisation of the classifiers to new sources. On the other hand, García-Díaz et al. (2022) and Russo et al. (2023) exclude the neutral class and work with a binary or multiclass Left–Right classifications of tweets from Spanish and Italian politicians respectively, but their work does not include longer texts. The binary classification might be justified as they worked with tweets, a genre where people tend to be more visceral and therefore probably more polarised. In our case, we need to be sure that the classifier generalises well to unseen sources and we stick to the 2-class task while minimising the number of neutral articles in training (see below).

**Distant Supervision.** As far as we know, only a manually annotated newspaper corpus in English (Baly et al., 2020) and another one in German (Aksenov et al., 2021) are available. We follow a different approach in the spirit of Kulkarni et al. (2018) and Kiesel et al. (2019). We do not manually annotate any article, but we trust All-Sides, MB/FC, Political Watch and Wikipedia (the latter only in cases where the information is not available in the previous sites) with their classification of a newspaper bias. We extract this information for newspapers from USA, Germany,

Spain and Catalonia. With the list of newspapers, their URL,[4] and their stance, we use OSCAR, a multilingual corpus obtained by filtering the Common Crawl (Ortiz Suárez et al., 2019; Abadji et al., 2021), to retrieve the articles. Appendix A lists the sources used in this work: 47 USA newspapers with 742,691 articles, 12 German with 143,200, 38 Spanish with 301,825 and 19 Catalan with 70,496.

**Topic Modelling.** Not all articles have a bias, some topics are more prone than others. While the Sports section of a newspaper is usually less prone to reflect political biases, the opposite happens with the International section. We therefore use topics to select a subset of relevant training data for our binary classification. We do topic modelling on the articles extracted from OSCAR using Mallet (McCallum, 2002) which applies LDA with Gibbs sampling. We cluster the data in both 10 and 15 groups per language, roughly corresponding to the number of sections a newspaper has. The keywords extracted for each topic are listed in Appendix B. We choose articles that fall under the topics we label as International, Government, Law & Justice, Economy, Live Science/Ecology, and specific language-dependent topics such as Immigration and Violence for English, Nazism for German, and Social for Spanish. The selection is done after the inspection of the keywords. For the final dataset, we do the union of the selected articles clustered to 10 and 15 topics. The process filters out 49% of the Spanish articles, 39% of the German and 31% of the English ones.

**Preprocessing and Cleaning.** We discard articles with more than 2000 or less than 20 words before cleaning. Afterwards, we remove headers, footers and any boilerplate text detected. This text has the potential to mislead a neural classifier, as it might encourage the classifier to learn to distinguish among newspapers rather than focusing on their political stance. We select a newspaper per language and stance for testing and clean manually their articles. To create a balanced training corpus for each language, we randomly select a similar number of Left and Right-oriented articles from the remaining collection. This balanced dataset is divided into training and validation as shown in Table 1 (top rows).

**ChatGPT/Bard Corpus.** We create a multilingual dataset with 101 articles. For this, we define

---

[4]This implies selecting all the articles that are under a domain name of a news outlet, whether they are news or not.

| | English (USA) | | German (Germany) | | Spanish (Spain) | | Catalan (Catalonia) | |
|---|---|---|---|---|---|---|---|---|
| | $\mathbb{L}$ | $\mathbb{R}$ | $\mathbb{L}$ | $\mathbb{R}$ | $\mathbb{L}$ | $\mathbb{R}$ | $\mathbb{L}$ | $\mathbb{R}$ |
| Training | 182,056 (756) | 178,463 (768) | 31,445 (550) | 30,745 (384) | 70,384 (874) | 67,888 (806) | – | – |
| Validation | 1,503 (723) | 1,497 (678) | 1,528 (570) | 1,472 (374) | 1,539 (878) | 1,461 (842) | – | – |
| Newspaper | 298 (731) | 413 (487) | 623 (278) | 276 (734) | 350 (844s) | 518 (731) | 2,105 (1,152) | 800 (538) |
| ChatGPTv02 | 101 (337) | | – | | 101 (346) | | – | |
| ChatGPTv03 | – | | 101 (299) | | – | | – | |
| ChatGPTv05 | 101 (585) | | 101 (436) | | 101 (553) | | 101 (496) | |
| ChatGPTv08 | 101 (v08a:470/v08b:467) | | 101 (v08a:321/v08b:324) | | 101 (v08a:454/v08b:464) | | 101 (v08a:378/v08b:365) | |
| Bardv08 | 101 (v08a:437/v08b:407) | | 101 (v08a:268/v08b:269) | | 101 (v08a:338/v08b:325) | | 101 (v08a:331/v08b:345) | |

Table 1: Number of articles (average word count in parentheses) divided as articles belonging to a newspaper with a Left ($\mathbb{L}$) and Right orientation ($\mathbb{R}$). For testing, we use newspapers not seen in training or validation: *Slate* ($\mathbb{L}$) and *The National Pulse* ($\mathbb{R}$) for USA, *My Heimat* ($\mathbb{L}$) and *die Preußische Allgemeine Zeitung* ($\mathbb{R}$) for Germany, *Mundo Obrero* ($\mathbb{L}$) and *El Diestro* ($\mathbb{R}$) for Spain and *Vilaweb* ($\mathbb{L}$) and *Diari de Tarragona* ($\mathbb{R}$) for Catalonia.

101 subjects including *housing prices*, *abortion*, *tobacco*, *Barak Obama*, etc. and translate them manually into the 4 languages (see Appendix D). The subjects consider topics prone to have a political stance such as those related to feminism, capitalism, ecologism, technology, etc. We also include proper names of people in the 4 countries being considered, whose biography may differ depending on the political stance of the writer. These subjects are inserted into the template prompt (and its translations into German, Spanish and Catalan):[5] *Write a newspaper article on [SUBJECT]$_{en}$*

We prompt ChatGPT (GPT-3.5-Turbo) five times using the same subjects across four time periods. We generate the dataset with ChatGPT versions of Feb 13 (v02), Mar 23 (v03), May 24 (v05) and Aug 3 (v08); we cover the 4 languages simultaneously only with the last two. ChatGPTv05 generates significantly longer texts than the other ones with an article-oriented structure with slots to be filled with the name of the author, date and/or city. Multilingual Bard was available later, and we prompt it twice during the same period as ChatGPTv8.[6] Table 1 shows the statistics for this corpus.

## 3 Political Stance Classification

**The Network.** We finetune XLM-RoBERTa large (Conneau et al., 2020), a multilingual transformer-based masked LM trained on 100 languages including the 4 we consider. The details of the network and the hyperparameter exploration per model are reported in Appendix F.

**The Models.** We train 4 models: 3 monolingual finetunings with the English, German and Spanish data, plus a multilingual one with the shuffled concatenation of the data. All models are based on multilingual embeddings (RoBERTa) finetuned either monolingually or multilingually. Notice that we do not train any model for Catalan. With this, we want to compare the performance of mono- and multilingual finetunings and explore the possibility of using multilingual models for zero-shot language transfer.

**Coarse Classification with Newspaper Articles.** Table 2 summarises the results. All the models achieve more than 95% accuracy on the validation set which is extracted from the same distribution as the training data. In order to see how the models behave with unseen data, we calculate the percentage of articles that are classified as Left ($\mathbb{L}$) and Right ($\mathbb{R}$) in the test newspapers of Table 1. We perform bootstrap resampling of the test sets with 1000 bootstraps to obtain confidence intervals at 95% level. We do not expect all the articles of a newspaper leaning towards the Left to *show clear characteristics* of the Left, but given that there is no neutral class, we expect the majority of them to *be classified as* Left. A good result is not necessarily 100%–0%, as this would not be realistic either. We consider that a newspaper has been classified as having a Left/Right political stance if more than 50% of its articles have been classified as such. These cases are boldfaced in Table 2.

This is the behaviour we obtain for all the test newspapers but for the German Right-oriented newspaper: die Preußische Allgemeine Zeitung (PAZ). The German model is trained only on 12

---

[5]More specific prompts did not lead to different styles for the first versions of ChatGPT, for the last one we added more information such as *...without subheaders.* to avoid excessive subsectioning and/or bullet points. Neither ChatGPT nor Bard did always follow properly the instruction. The dataset we provide includes the prompts we used.

[6]Prompted 14–21 August 2023 from Berlin for English and German and from Barcelona for Spanish and Catalan as, contrary to ChatGPT, the generation depends on the location.

| | English | | | | German | | | | Spanish | | | | Catalan | |
| | Monolingual | | Multilingual | | Monolingual | | Multilingual | | Monolingual | | Multilingual | | Multilingual | |
| | $\mathbb{L}$ | $\mathbb{R}$ | $\mathbb{L}$ | $\mathbb{R}$ | $\mathbb{L}$ | $\mathbb{R}$ | $\mathbb{L}$ | $\mathbb{R}$ | $\mathbb{L}$ | $\mathbb{R}$ | $\mathbb{L}$ | $\mathbb{R}$ | $\mathbb{L}$ | $\mathbb{R}$ |
|---|---|---|---|---|---|---|---|---|---|---|---|---|---|---|
| Val. Acc (%) | 97.9 | | 96.9 | | 99.2 | | 96.9 | | 95.9 | | 96.9 | | — | |
| | Classification (% of articles per stance) | | | | | | | | | | | | | |
| Newspaper $\mathbb{L}$ | **82**±5 | 18±4 | **81**±5 | 19±4 | **87**±3 | 13±2 | **65**±4 | 35±4 | **55**±5 | 45±5 | **61**±5 | 39±5 | **65**±2 | 35±2 |
| Newspaper $\mathbb{R}$ | 11±3 | **89**±3 | 7±2 | **93**±3 | **71**±6 | 29±6 | **65**±6 | 35±5 | 12±3 | **88**±3 | 19±3 | **81**±4 | 13±2 | **87**±2 |
| ChatGPTv02 | **75**±9 | 25±8 | **93**±5 | 7±5 | – | – | – | – | **65**±10 | 35±10 | **53**±10 | 47±10 | – | – |
| ChatGPTv03 | – | – | – | – | **97**±4 | 3±3 | **69**±9 | 31±9 | – | – | – | – | – | – |
| ChatGPTv05 | 26±9 | **74**±9 | 40±9 | **60**±9 | **96**±5 | 4±3 | **65**±9 | 35±9 | 25±9 | **75**±9 | 26±9 | **74**±8 | **71**±9 | 29±9 |
| ChatGPTv08a | **54**±10 | 46±10 | **85**±8 | 15±6 | **99**±3 | 1±1 | **100**±2 | 0±0 | 50±10 | 50±10 | 40±10 | **60**±10 | 50±10 | 50±9 |
| ChatGPTv08b | **52**±10 | 48±10 | **85**±8 | 15±6 | **100**±2 | 0±0 | **100**±2 | 0±0 | **51**±10 | 49±10 | 36±10 | **64**±9 | 47±10 | **53**±10 |
| Bardv08a | **57**±11 | 43±10 | **75**±9 | 25±8 | **82**±8 | 18±7 | **82**±8 | 18±7 | **74**±9 | 26±8 | 35±9 | **65**±9 | **66**±9 | 34±9 |
| Bardv08b | **61**±10 | 39±10 | **82**±8 | 18±7 | **81**±8 | 19±7 | **90**±7 | 10±5 | **74**±9 | 26±8 | 44±10 | **56**±10 | **68**±9 | 32±9 |

Table 2: (top) Accuracy of the 4 finetuned models on the corresponding validation sets. (bottom) Percentage of articles classified as having a Left ($\mathbb{L}$) and a Right ($\mathbb{R}$) orientation (columns) for the test newspapers and the Bard/ChatGPT generated articles at four different time periods (rows). The majority stance is boldfaced.

newspapers to be compared to the 47 in English and 38 in Spanish. The incorrect classification might be an indication that diversity is a key aspect for the final model performance. Multilinguality does not help and 65% of the PAZ articles are still classified as Left oriented. We also assess the effectiveness of the English model on the German data, two close languages. We acknowledge that the topics of the USA and German newspapers might differ a lot, but the high diversity of the English training data could potentially compensate for this. The English model is able to correctly classify the German My Heimat as a Left-oriented newspaper ($\mathbb{L}$: 67±3%) and PAZ as a Right-oriented one ($\mathbb{R}$: 58±5%). We again attribute the difference to the German model being trained on a corpus lacking diversity. When we use the multilingual system, the dominant factor distinguishing the outputs is the language itself rather than the stance. The addition of English data is insufficient to alter the classification significantly. When we use the English system, the language does not play a role any more and only the stance features are considered. When we apply the English model to the Catalan newspapers we do not obtain satisfactory results though (95±1% for the Left but 16±3% for the Right newspaper) showing that the relatedness across languages is important. The multilingual model however properly detects the stance of the Catalan newspapers probably because it has been trained with an heterogeneous corpus that includes a related language (Spanish). We are able to perform zero-shot language transfer classification when we deal with close related languages.

**Coarse Classification with ILM-generated Articles.** The bottom part of Table 2 details the results.

We first focus on the English and Spanish models as the German one did not properly classify our test newspapers. The most relevant aspect to notice in **ChatGPT** is the strong change in political stance between February (v02) and May (v05) followed by a movement towards neutrality in August (v08). We checked that this polarity change is not an effect of the length of the outputs —the major shallow change in the generated articles. The training data in English has 5,730 $\mathbb{L}$–6,988 $\mathbb{R}$ articles with $584 < length(words) < 624$ (similar to ChatGPTv05 length) and 4,563 $\mathbb{L}$–7,127 $\mathbb{R}$ articles with $331 < length < 371$ (similar to ChatGPTv02). In both cases the number of articles is larger for the Right stance, but the prediction for ChatGPTv02 clearly points towards the Left, rejecting the hypothesis that length plays a role in the classification. A similar thing happens for Spanish. According to our models, the May 24th version of ChatGPT in English and Spanish would have an editorial line close to the Right ideology, which differs from the ideology of the previous versions. Notably, this period corresponds to the time when ChatGPT experienced a performance drop in several tasks according to Chen et al. (2023). The German and Catalan outputs would still show an imprint from the Left ideology also in v05 but more diverse training data would be needed to confirm this with our monolingual models. It is interesting to notice that if we use the English monolingual model for German and Catalan, we still get the Left imprint (60±10% for German and 87±7% for Catalan). So we have indications that the political stance of ChatGPT depends on the language, which is not surprising in a data-driven system. The last

version, ChatGPTv08, produces the most neutral texts, with only German clearly leaning towards the Left. The two generations, v08a and v08b, show that results are robust and are not tied to a particular generation.

There is only a version available for multilingual **Bard** that covers our time frame.[7] The variation between generations is larger for Bard than for ChatGPT but, comparing v08 versions, Bard points towards the Left in a more consistent way across languages. Bard's political orientation can also be determined by its answers to political test or quiz questions. The Political Compass (PC) site[8] defines 62 propositions to identify the political ideology —with an European/Western view— in two axes: economic policy (Left–Right) and social policy (Authoritarian–Libertarian), both in the range [-10,10]. Each proposition is followed by 4 alternatives: strongly agree, agree, disagree and strongly disagree. When prompted with the questionnaire,[9] Bard's scores are (-6.50, -4.77) for English, (-8.00, -7.13) for German, (-5.75, -4.15) for Spanish and (-6.75, -4.56) for Catalan, where the first number corresponds to the economic policy and the second to the social policy. The results are in concordance with Table 2 and give an indirect validation of our method which does not rely on direct questions.[10]

This kind of analysis is not possible with Chat-GPT any more as it refrains from expressing opinions and preferences, demonstrating the relevance of an approach that detects the leaning in a more indirect way. Also notice that these questionnaires are well-known and public, so it would be easy to instruct a LM to avoid the questions or react to its propositions in a neutral manner. Previous work used only political tests and questionnaires to estimate ChatGPT's orientation. Hartmann et al. (2023) used PC, 38 political statements from the voting advice application Wahl-O-Mat (Germany) and 30 from StemWijzer (the Netherlands) to conclude that ChatGPT's ideology in its version of Dec 15 2022 was pro-environmental and left-libertarian.

A study conducted by the Manhattan Institute for Policy Research[11] reported that ChatGPT tended to give responses typical of Left-of-center political viewpoints for English (Rozado, 2023). The authors administered 15 political orientation tests to the ChatGPT version of Jan 9. Their results are consistent with our evaluation of the Feb 13 model. Finally, Motoki et al. (2023) performed a battery of tests based on PC to show that ChatGPT is strongly biased towards the Left. The authors do not state the version they use, but the work was submitted on March 2023. All these results are therefore before the move to the Right we detected in May.

## 4 Summary and Conclusions

Media sources have an editorial line and an associated bias. Getting rid of political biases is difficult for humans, but being aware of them helps us getting a global view of news. Biases are sometimes clear and/or appear in form of harmful text, but sometimes are subtle and difficult to detect. These subtle hidden biases are potentially dangerous and lead to manipulation whenever we are not aware of them. In this work, we systematically studied the subtle political biases behind ChatGPT and Bard, those that appear without assigning any persona role (Deshpande et al., 2023). We showed that ChatGPT's orientation changes with time and it is different across languages. Between Feb and Aug 2023, ChatGPT transitioned from a Left to Neutral political orientation, with a Right-leaning period in the middle for English and Spanish. The evolution for Bard cannot be studied yet. Its current version as of Aug 2023 consistently shows Left-leaning for the 4 languages under study. This bias is independent on the factual mistakes that the model generates, and should also be considered by its users. We provide models to regularly check the bias in text generations for USA, Germany and Spain, as well as in closely related political contexts and languages using a zero-shot approach.

As a by-product of our analysis, we created a multilingual corpus of 1.2M newspaper articles with coarse annotations of political stance and topic. We show that distant supervision allows us to build meaningful models for coarse political stance classification as long as the corpus is diverse. We make available this data together with the LMs generations and our code through Zenodo (España-Bonet, 2023) and Github.[12]

---

[7]Notice that the version we use does not officially support Catalan, but native speakers confirmed that generations are mostly correct and fluent with few grammatical mistakes.

[8]https://www.politicalcompass.org/test (accessed between 13th and 20th August 2023)

[9]The Spanish questionnaire was translated into Catalan, as the questionnaire was not available.

[10]Even though, similarly to people, it is possible for an ILM to *say* one thing (chose an option for a proposition) and *act* (write a text) in an inconsistent way.

[11]A conservative think tank according to Wikipedia.

[12]https://github.com/cristinae/docTransformer

## Limitations

We are assuming that *All media sources have an editorial line and an associated bias*, and we treat the ILM as any other media source. We do not consider the possibility of a ChatGPT or Bard article being unbiased. This is related to the distant supervision method used to gather the data that currently allows for a binary political stance annotation. Since manually annotating hundreds of thousands of articles with political biases in a truly multilingual setting seems not possible in the foreseeable future, we decided to implement a completely data-based method and study its language and culture transfer capabilities.

Using distant supervision for detecting the political stance at article level is a delicate topic though. First, because the same newspaper can change ideology over time. Second, and this is more related to the content of an individual article, non-controversial subjects might not have a bias. Even in cases where bias exists, there is a spectrum ranging from the extreme Left to the extreme Right, rather than a clear-cut division between the two ideologies.

In order to quantify and if possible mitigate the current limitations, we plan to conduct a stylistic analysis of the human-annotated corpora (Baly et al., 2020; Aksenov et al., 2021) and compare it to our semi-automatically annotated corpus. As a follow-up of this work, we will perform a stylistic analysis of the ILM-generated texts too as a similar style between the training data and these texts is needed to ensure good generalisation and transfer capabilities.

## Ethics Statement

We use generative language models, ChatGPT and Bard, to create our test data. Since we deal with several controversial subjects (death penalty, sexual harassment, drugs, etc.) the automatic generation might produce harmful text. The data presented here has not undergone any human revision. We analyse and provide the corpus as it was generated, along with the indication of the systems version used.

## Acknowledgements

The author thanks the anonymous reviewers for insightful comments and discussion. Eran dos ifs.

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

# A  Newspapers in OSCAR 22.01

| Language | Leaning | Newspaper | URL | #Articles | Word average |
|---|---|---|---|---|---|
| *en* | 𝕃 | ABC News | abcnews.go.com | 28101 | 659 |
| *en* | 𝕃 | AlterNet | www.alternet.org | 1228 | 1680 |
| *en* | 𝕃 | Associated Press News | apnews.com | 47090 | 679 |
| *en* | 𝕃 | Axios | www.axios.com | 2119 | 733 |
| *en* | 𝕃 | Buzzfeed News | www.buzzfeednews.com | 11025 | 885 |
| *en* | 𝕃 | CBS News | www.cbsnews.com | 19761 | 534 |
| *en* | 𝕃 | CNN | www.cnn.com | 15273 | 712 |
| *en* | 𝕃 | | edition.cnn.com | 19204 | 772 |
| *en* | 𝕃 | Democracy Now! | www.democracynow.org | 1481 | 1213 |
| *en* | 𝕃 | HuffPost | www.huffpost.com | 45721 | 961 |
| *en* | 𝕃 | | preview.www.huffpost.com | 181 | 1097 |
| *en* | 𝕃 | | staging.www.huffpost.com | 266 | 1142 |
| *en* | 𝕃 | Insider | www.insider.com | 3705 | 1047 |
| *en* | 𝕃 | Mother Jones | www.motherjones.com | 12332 | 688 |
| *en* | 𝕃 | MSNBC News | www.msnbc.com | 20682 | 250 |
| *en* | 𝕃 | NBC News | www.nbcnews.com | 38620 | 518 |
| *en* | 𝕃 | NPR | www.npr.org | 45517 | 793 |
| *en* | 𝕃 | | archive.nytimes.com | 524 | 913 |
| *en* | 𝕃 | Politico | www.politico.com | 30842 | 804 |
| *en* | 𝕃 | Slate | www.slate.com | 674 | 761 |
| *en* | 𝕃 | The Atlantic | www.theatlantic.com | 41816 | 750 |
| *en* | 𝕃 | The Daily Beast | www.thedailybeast.com | 27771 | 806 |
| *en* | 𝕃 | The Economist | www.economist.com | 38672 | 822 |
| *en* | 𝕃 | The Intercept | theintercept.com | 17766 | 1328 |
| *en* | 𝕃 | The New Yorker | www.newyorker.com | 18698 | 688 |
| *en* | 𝕃 | The New York Times | www.nytimes.com | 2382 | 446 |
| *en* | 𝕃 | The Washington Post | www.washingtonpost.com | 11944 | 1009 |
| *en* | 𝕃 | USA Today | www.usatoday.com | 4526 | 1308 |
| *en* | 𝕃 | Vox | www.vox.com | 6746 | 934 |
| *en* | ℝ | American Greatness | amgreatness.com | 2405 | 849 |
| *en* | ℝ | American Thinker | www.americanthinker.com | 9891 | 805 |
| *en* | ℝ | Breitbart News Network | www.breitbart.com | 18762 | 651 |
| *en* | ℝ | ConservativeHQ | www.conservativehq.com | 1685 | 455 |
| *en* | ℝ | Fox News | www.foxnews.com | 25074 | 770 |
| *en* | ℝ | InfoWars | www.infowars.com | 127 | 574 |
| *en* | ℝ | Life Action | www.liveaction.org | 3095 | 740 |
| *en* | ℝ | National Review | www.nationalreview.com | 7251 | 946 |
| *en* | ℝ | Reason | reason.com | 23553 | 489 |
| *en* | ℝ | The American Conservative | www.theamericanconservative | 1228 | 935 |
| *en* | ℝ | The Blaze | www.theblaze.com | 34 | 1291 |
| *en* | ℝ | The Daily Caller | dailycaller.com | 26125 | 449 |
| *en* | ℝ | | amp.dailycaller.com | 61 | 540 |
| *en* | ℝ | The Daily Wire | www.dailywire.com | 7143 | 603 |
| *en* | ℝ | The Epoch Times | www.theepochtimes.com | 27553 | 695 |
| *en* | ℝ | The Federalist | thefederalist.com | 7759 | 835 |
| *en* | ℝ | The Gateway Pundit | www.thegatewaypundit.com | 5232 | 431 |
| *en* | ℝ | The National Pulse | thenationalpulse.com | 481 | 593 |
| *en* | ℝ | The Washington Free Beacon | freebeacon.com | 7310 | 593 |
| *en* | ℝ | The Washington Times | www.washingtontimes.com | 19486 | 710 |
| *en* | ℝ | The Spectator | spectator.org | 14903 | 762 |
| *en* | ℝ | Washington Examiner | m.washingtonexaminer.com | 30 | 647 |
| *en* | ℝ | WND | www.wnd.com | 21099 | 664 |
| *de* | 𝕃 | Die Zeit | www.zeit.de | 12 | 4211 |
| *de* | 𝕃 | Die Tageszeitung | taz.de | 40253 | 694 |
| *de* | 𝕃 | | blogs.taz.de | 1336 | 723 |
| *de* | 𝕃 | | genossenschaft.taz.de | 16 | 482 |
| *de* | 𝕃 | DW News | www.dw.com | 2290 | 1075 |
| *de* | 𝕃 | Junge Welt | www.jungewelt.de | 7386 | 279 |
| *de* | 𝕃 | My Heimat | www.staz.de | 2499 | 329 |
| *de* | 𝕃 | Neues Deutschland | www.nd-aktuell.de | 6608 | 785 |
| *de* | 𝕃 | Süddeutsche Zeitung | www.sueddeutsche.de | 31578 | 583 |
| *de* | ℝ | Bild | www.bild.de | 24459 | 279 |
| *de* | ℝ | Frankfurter Allgemeine Zeitung | www.faz.net | 12340 | 478 |

| Language | Leaning | Newspaper | URL | #Articles | Word average |
|---|---|---|---|---|---|
| *de* | ℝ | Die Welt | www.welt.de | 9546 | 553 |
| *de* | ℝ | Junge Freiheit | jungefreiheit.de | 8379 | 496 |
| *de* | ℝ | Preußische Allgemeine Zeitung | paz.de | 385 | 945 |
| *es* | 𝕃 | Cuarto poder | www.cuartopoder.es | 3105 | 1353 |
| *es* | 𝕃 | De Verdad digital | deverdaddigital.com | 3422 | 1071 |
| *es* | 𝕃 | Diario progresista | www.diarioprogresista.es | 217 | 599 |
| *es* | 𝕃 | Digital Sevilla | digitalsevilla.com | 751 | 494 |
| *es* | 𝕃 | elcomunista.net | elcomunista.net | 736 | 1974 |
| *es* | 𝕃 | elDiario.es | www.eldiario.es | 17846 | 1887 |
| *es* | 𝕃 | El Obrero | elobrero.es | 2856 | 1094 |
| *es* | 𝕃 | El país | elpais.com | 49923 | 903 |
| *es* | 𝕃 | El periódico | www.elperiodico.com | 26940 | 647 |
| *es* | 𝕃 | El plural | www.elplural.com | 570 | 759 |
| *es* | 𝕃 | El siglo de Europa | elsiglodeuropa.es | 207 | 1596 |
| *es* | 𝕃 | HuffPost | www.huffingtonpost.es | 9967 | 1097 |
| *es* | 𝕃 | La República | larepublica.es | 125 | 842 |
| *es* | 𝕃 | Los Replicantes | www.losreplicantes.com | 5398 | 735 |
| *es* | 𝕃 | Mundiario | www.mundiario.com | 16299 | 461 |
| *es* | 𝕃 | Mundo Obrero | www.mundoobrero.es | 527 | 764 |
| *es* | 𝕃 | Postdigital | postdigital.es | 209 | 824 |
| *es* | 𝕃 | Público | www.publico.es | 9126 | 1097 |
| *es* | ℝ | Adelante España | adelanteespana.com | 178 | 731 |
| *es* | ℝ | Altavoz de sucesos | altavozdesucesos.es | 136 | 257 |
| *es* | ℝ | Disidentia | disidentia.com | 167 | 967 |
| *es* | ℝ | El Confidencial | www.elconfidencial.com | 23560 | 1037 |
| *es* | ℝ | El correo de Andalucía | elcorreoweb.es | 4940 | 539 |
| *es* | ℝ | El correo de España | elcorreodeespana.com | 1896 | 1343 |
| *es* | ℝ | El diestro | www.eldiestro.es | 3046 | 1321 |
| *es* | ℝ | El Imparcial | www.elimparcial.es | 4777 | 518 |
| *es* | ℝ | El independiente | www.elindependiente.com | 7061 | 806 |
| *es* | ℝ | El Mundo | www.elmundo.es | 24828 | 901 |
| *es* | ℝ | Hispanidad | www.hispanidad.com | 2021 | 1414 |
| *es* | ℝ | Información | www.informacion.es | 1432 | 768 |
| *es* | ℝ | La Gaceta | gaceta.es | 326 | 986 |
| *es* | ℝ | La Razón | www.larazon.es | 22283 | 619 |
| *es* | ℝ | La Vanguardia | www.lavanguardia.com | 26533 | 777 |
| *es* | ℝ | La voz de Galicia | www.lavozdegalicia.es | 6709 | 935 |
| *es* | ℝ | Libertad digital | www.libertaddigital.com | 12591 | 561 |
| *es* | ℝ | OK Diario | okdiario.com | 1636 | 780 |
| *es* | ℝ | Periodista digital | www.periodistadigital.com | 9386 | 1690 |
| *es* | ℝ | Voz libre | vozlibre.com | 329 | 1441 |
| *ca* | | Ara | www.ara.cat | 14317 | 664 |
| *ca* | | Ara Balears | www.arabalears.cat | 3743 | 626 |
| *ca* | | CatalunyaPress | www.catalunyapress.cat | 2669 | 411 |
| *ca* | | Crític | www.elcritic.cat | 447 | 1516 |
| *ca* | | Diari d'Andorra | www.diariandorra.ad | 5036 | 444 |
| *ca* | | Diari de Girona | www.diaridegirona.cat | 4706 | 456 |
| *ca* | | Diari Segre | www.segre.com | 1196 | 945 |
| *ca* | | El 9 Nou | el9nou.cat | 287 | 684 |
| *ca* | | | 9club.el9nou.cat | 11 | 334 |
| *ca* | | El Nacional | www.elnacional.cat | 1400 | 575 |
| *ca* | | El Punt Avui | www.elpuntavui.cat | 11571 | 666 |
| *ca* | | e-Notícies | www.e-noticies.cat | 20 | 240 |
| *ca* | | La República | www.larepublica.cat | 1137 | 662 |
| *ca* | | La Nació digital | www.naciodigital.cat | 14183 | 602 |
| *ca* | | Regió 7 | www.regio7.cat | 2728 | 476 |
| *ca* | 𝕃 | El Periodico | www.elperiodico.cat | 3095 | 792 |
| *ca* | 𝕃 | jornal.cat | www.jornal.cat | 106 | 648 |
| *ca* | 𝕃 | Público | www.publico.es | 85 | 1745 |
| *ca* | 𝕃 | VilaWeb | www.vilaweb.cat | 2955 | 2247 |
| *ca* | ℝ | Diari de Tarragona | www.diaridetarragona.com | 800 | 540 |

Table 3: List of newspapers available in the OSCAR corpus (version 22.01) for the four languages used in this work.

# B  Topics

| ID | % | Label | Keywords |
|---|---|---|---|
| **10:0** | 0.12 | economy | percent year government tax money billion economic economy years pay financial federal workers u.s jobs people companies market business spending energy plan climate prices work policy trade bank time budget |
| 10:1 | 0.16 | culture | time people life years work book story love family good day film music man feel women great told young year lot movie kind long art friends woman times thought york |
| **10:2** | 0.10 | international | u.s military government war president united security china people minister country foreign israel iran iraq international american countries russia chinese officials forces nuclear political attack news north party intercept group |
| **10:3** | 0.15 | government | trump president house democrats campaign republican obama election senate democratic republicans party white vote clinton political biden voters presidential donald news people time congress conservative sen gop washington support national |
| 10:4 | 0.08 | technology | company companies business facebook data technology online market news media internet work google social people users time digital twitter service tech site products including content industry ceo free customers apple |
| **10:5** | 0.13 | law & justice | court law federal case department u.s justice investigation government attorney told report legal news officials public judge office fbi security administration general supreme trump border president statement house criminal evidence |
| 10:6 | 0.13 | education | people school students women black american education rights children university america schools public white years political work religious social community americans college time life men country church history parents free |
| 10:7 | 0.08 | covid | health people covid coronavirus care medical vaccine pandemic share cases children patients news abortion virus disease study drug public hospital time percent risk email women doctors university cancer deaths treatment |
| 10:8 | 0.10 | hotchpotch I | police game city team time gun told shooting season people year-old year man officers games shot night players killed news sports day left officer football sunday death week county play |
| 10:9 | 0.09 | hotchpotch II | water food people city years space year air time area day climate local national miles travel flight land small island oil south park coast scientists north long sea california change |
| 15:0 | 0.09 | education | school people students women children health university work study time education parents schools life kids percent college years care medical child high family student young cancer good feel day patients |
| **15:1** | 0.04 | immigration | immigration border u.s people immigrants mexico government illegal country united president american marijuana drug migrants children year years security work democracy administration america news mexican texas refugees legal today enforcement |
| **15:2** | 0.07 | government I | trump news president u.s intelligence media report house investigation security fbi department government white intercept campaign officials committee clinton told story national trump's justice washington russian administration member documents donald |
| 15:3 | 0.14 | hotchpotch | time people life years love film book story work music family day good year movie man art great women told young kind feel lot york series long woman books night |
| 15:4 | 0.06 | covid | covid health people coronavirus pandemic vaccine share cases virus york news public medical care officials u.s week vaccinated disease workers deaths vaccines reported hospital city twitter email patients told facebook |
| **15:5** | 0.11 | violence | police people told city news death man officers gun county shooting family year-old black killed violence officer case video school reported charges prison department crime years shot arrested time authorities |
| **15:6** | 0.07 | international I | china chinese government countries minister north european united u.s south party country international president russia korea foreign europe prime russian british political trade germany year years french leader global britain |
| **15:7** | 0.13 | government II | trump president house democrats republican campaign obama senate election democratic republicans party vote biden voters white presidential political clinton sen gop percent congress support donald candidate people time washington national |
| **15:8** | 0.13 | hotchpotch | people american political america president black time years white war conservative media fact americans left history conservatives movement good country church power speech book social public free life religious man |
| **15:9** | 0.10 | economy | percent tax year money government billion economic economy pay financial years federal jobs workers people spending market companies business prices bank u.s budget insurance rate plan debt health growth costs |
| **15:10** | 0.08 | ecologism | water climate food years energy space people city year oil change time environmental air gas area power national natural miles day land scientists small science local emissions carbon earth weather |
| **15:11** | 0.05 | international II | u.s military war israel iran iraq government security forces afghanistan united syria attack people president israeli islamic attacks killed american troops nuclear army obama country muslim group terrorist officials weapons |
| 15:12 | 0.03 | sports | game team season games players sports football play time win coach league year nfl points player won teams left field week good fans yards sunday night final big college played |
| **15:13** | 0.09 | law & justice | court law federal rights case supreme legal abortion justice public government judge decision laws action u.s amendment department order attorney lawsuit news ban cases ruling issue general filed texas policy |
| 15:14 | 0.07 | technology | company companies business facebook technology market data online internet google media users products digital work tech people service industry news site time social content customers ceo free apple firm year |

Table 4: Topics (with 10 and 15 clusters) obtained with Mallet on OSCAR's English newspaper documents. Clusters boldfaced and colored in blue are used to build the training data.

| ID | % | Label | Keywords |
|---|---|---|---|
| 10:0 | 0.14 | culture | film leben frau bild welt musik sehen erzählt frauen mutter kunst geschichte berlin paar vater weiß familie liebe steht künstler zeigt bühne kinder leute spielt bilder eher publikum männer band |
| **10:1** | 0.15 | hotchpotch | welt deutschen deutschland gesellschaft buch leben geschichte deutsche beitrag frage frauen politik junge politischen politische berlin medien kirche juden thema wissen krieg freiheit steht eher leute kultur staat arbeit sprache |
| 10:2 | 0.11 | law & justice | polizei gericht fall polizisten laut staatsanwaltschaft opfer deutschland bild berlin gewalt frau täter jährige richter ermittlungen urteil verletzt prozess behörden foto personen dpa verfahren angaben verurteilt offenbar beamten berliner männer |
| **10:3** | 0.11 | government | spd partei cdu grünen merkel afd berlin deutschland fdp union politik bundestag linke csu parteien wahl grüne koalition regierung angela frage linken bundesregierung stimmen kanzlerin berliner mehrheit müsse kritik steht |
| **10:4** | 0.10 | economy | euro millionen unternehmen deutschland milliarden geld deutschen wirtschaft deutsche zahlen berlin kosten dollar bank banken welt kunden europa folgen zukunft europäischen arbeit laut usa china land firmen bild konzern markt |
| 10:5 | 0.05 | sport | bayern spiel trainer spieler fußball spielen saison bild deutschen mannschaft fans platz spiele team sieg sport tor münchen league liga deutsche minuten letzten dortmund steht bundesliga verein ball teilen datensicherheit |
| 10:6 | 0.07 | technology | daten inhalte finden informationen internet facebook vergleich inhalt google twitter zustimmung personenbezogene artikel nutzer angezeigt übermittelt einverstanden brauchen laden gerät anzeige forscher netz sozialen bild unternehmen euro kunden produkte app |
| 10:7 | 0.09 | covid | kinder deutschland frauen eltern zahl schulen schule leben pandemie schüler patienten laut kindern arbeit woche liegt bayern berlin zahlen kind corona arbeiten bekommen deutschen flüchtlinge studie gilt personen wochen millionen |
| **10:8** | 0.10 | international | regierung usa land trump präsident türkei russland europa staaten deutschland soldaten china israel präsidenten syrien afghanistan landes welt krieg iran flüchtlinge frankreich grenze ausland donald millionen russischen hauptstadt armee obama |
| 10:9 | 0.10 | local | stadt euro wasser münchen meter straße steht ort kilometer stehen haus leben liegt quelle auto platz fahren berlin augsburg gebäude münchner bild straßen tiere besucher paar unterwegs projekt millionen sieht |
| 15:0 | 0.05 | covid | bayern pandemie corona münchen landkreis deutschland augsburg oktober zahl junge freitag woche virus montag welt covid coronavirus mittwoch stadt patienten november dienstag donnerstag liegt maßnahmen sonntag bayerischen geimpft feiern wochen |
| 15:1 | 0.06 | local | stadt euro vergleich berlin auto wohnungen straße autos bahn münchen fahren meter gebäude hamburg quelle bau platz kilometer berliner projekt haus finden kosten straßen steht stehen gebaut millionen wohnen bauen |
| 15:2 | 0.08 | social | frauen kinder deutschland eltern schule arbeit berlin schulen schüler leben männer kindern arbeiten kind frau flüchtlinge studie deutschen bekommen universität lehrer zahl thema familien familie jungen laut berliner euro stellen |
| **15:3** | 0.11 | government | spd cdu grünen partei merkel afd berlin fdp deutschland union bundestag csu grüne linke koalition parteien politik wahl angela bild kanzlerin regierung bundesregierung geben berliner linken kritik müsse seehofer dpa |
| **15:4** | 0.06 | live science | wasser tiere deutschland forscher natur erde wissenschaftler millionen grad meter welt leben klimawandel studie pflanzen wald liegt landwirtschaft umwelt essen weltweit fleisch tonnen bauern tier kilometer körper bild gesundheit laut |
| **15:5** | 0.06 | Nazism | deutschen berlin geschichte kirche juden buch ddr deutsche berliner seite deutschland leben krieg bücher jüdischen museum stadt tod ausstellung papst weltkrieg kultur hitler nazis jüdische literatur polen verlag schriftsteller welt |
| 15:6 | 0.18 | family | leben frau paar leute bild weiß familie steht kinder erzählt mutter welt vater frauen sehen männer junge eltern geld wissen sieht stehen bisschen liebe haus kind kopf hause schnell jährige |
| 15:7 | 0.19 | hotchpotch | welt deutschland frage gesellschaft politik beitrag deutschen politischen politische leben eher wissen buch fragen deutsche freiheit art europa staat medien steht leute thema problem geschichte sehen rolle demokratie all klar |
| 15:8 | 0.05 | sport | bayern trainer spiel spieler fußball spielen saison mannschaft deutschen fans platz team sieg sport bild spiele tor liga league deutsche minuten münchen dortmund letzten bundesliga verein ball stadion steht minute |
| **15:9** | 0.12 | law & justice | polizei gericht fall polizisten staatsanwaltschaft laut opfer täter richter jährige ermittlungen urteil gewalt bild verletzt berlin prozess deutschland behörden verfahren dpa personen angaben foto frau verurteilt beamten offenbar hamburg montag |
| 15:10 | 0.07 | culture | film musik kunst künstler welt berlin band bühne sehen geschichte bild filme publikum zeigt spielt ausstellung theater bilder album regisseur leben roman art schauspieler kino musiker berliner york werk erzählt |
| 15:11 | 0.06 | technology | daten inhalte artikel facebook internet inhalt informationen finden twitter google medien netz bild zustimmung personenbezogene laden nutzer angezeigt einverstanden übermittelt sozialen zeitung wahrheit digitalen journalismus brauchen soziale gerät app unternehmen |
| **15:12** | 0.07 | international I | trump usa regierung präsident land china donald partei präsidenten obama frankreich staaten wahl europa washington parlament großbritannien us-präsident europäischen brüssel trumps stimmen welt landes italien york biden macron london mehrheit |
| **15:13** | 0.10 | economy | euro millionen unternehmen deutschland milliarden geld deutschen wirtschaft deutsche zahlen dollar kosten bank banken kunden berlin europäischen europa länder bundesregierung griechenland firmen konzern folgen laut krise mitarbeiter insgesamt land markt |
| **15:14** | 0.07 | international II | türkei russland regierung deutschland land israel flüchtlinge syrien soldaten afghanistan iran usa präsident krieg grenze europa bundeswehr russischen putin ukraine armee türkischen deutsche irak türkische staaten taliban moskau russische stadt |

Table 5: Topics (with 10 and 15 clusters) obtained with Mallet on OSCAR's German newspaper documents. Clusters colored in blue are used to build the training data.

| ID | % | Label | Keywords |
|---|---|---|---|
| 10:0 | 0.14 | hotchpotch | mensaje españa sociedad años opinión vida mundo denunciar política mujeres gente personas historia privado educación social artículo iglesia redactar realidad país citar políticos forma enviar libertad derecho españoles leer papa |
| **10:1** | 0.16 | economy | millones euros españa año economía empresas años gobierno crisis mercado sector empresa país banco social países económica información compañía dinero política europea precio trabajadores sistema caso deuda servicios mes meses |
| 10:2 | 0.16 | culture | años vida mundo historia casa año película cine libro obra mujer familia música serie madre gente padre the arte programa españa premio televisión director hijo novela amor hombre joven noche |
| **10:3** | 0.12 | government | gobierno sánchez psoe partido presidente elecciones congreso política pedro iglesias ciudadanos españa rajoy madrid pablo cataluña vox partidos electoral líder ley socialista moncloa díaz izquierda político votos diputados constitución euros |
| 10:4 | 0.15 | local | barcelona madrid ciudad años centro ayuntamiento covid catalunya personas españa zona visto comunidad periódico año calle metros relacionadas noticias local proyecto palma galicia vecinos hora plaza euros comentado nacional semana |
| 10:5 | 0.11 | science | años forma personas salud mundo tipo estudio tecnología vida productos datos sistema información agua explica internet caso usuarios año investigación importante calidad universidad mejores permite cambio consumo españa problema enfermedad |
| 10:6 | 0.06 | covid | españa coronavirus casos sociedad covid mapas evolución gobierno datos vacunación gráficos política mundo personas socios canarias madrid leído años pandemia contagios sanidad variante ómicron salud avanza hazte enlace copiar vacuna |
| **10:7** | 0.11 | law & justice | años caso madrid policía comentarios juez tribunal vox público españa díaz hombre sociedad ayuso justicia ley fiscalía investigación comunidad casado nacional judicial yolanda sentencia delito publicidad prisión audiencia civil noticias |
| 10:8 | 0.07 | sport | madrid real partido equipo fútbol años españa club liga temporada barcelona jugador jugadores barça balón año juego carrera mundial puntos partidos minutos jornada messi español gol jugar historia campo entrenador |
| **10:9** | 0.10 | international | país presidente gobierno años países unidos internacional guerra trump china eeuu personas ministro europa rusia información mundo diciembre seguridad méxico política militar nacional noviembre francia europea elecciones actualidad millones frente |
| 15:0 | 0.07 | live science | salud años estudio personas agua vida enfermedad forma investigación pacientes cáncer virus hospital tipo riesgo alimentos explica productos tratamiento enfermedades científicos mundo caso animales año médicos consumo médico investigadores niños |
| 15:1 | 0.05 | Catalonia | barcelona catalunya generalitat catalán covid periódico cataluña puigdemont catalana relacionadas noticias visto comentado temas mossos pasaporte lee sant govern minutos erc años confiar jordi pandemia centro lingüística parlament coronavirus directo |
| **15:2** | 0.14 | government I | gobierno partido psoe presidente sánchez elecciones política españa congreso ciudadanos madrid rajoy ley partidos electoral pedro líder izquierda votos vox político socialista pablo país diputados ejecutivo debate iglesias portavoz comunidad |
| **15:3** | 0.11 | law & justice | años caso policía juez comentarios madrid tribunal público sociedad justicia investigación fiscalía nacional españa comunidad euros judicial ley civil sentencia delito prisión audiencia ayuso mujer fiscal recuerda juicio guardia juzgado |
| 15:4 | 0.05 | covid | españa coronavirus casos sociedad covid mapas evolución gobierno datos vacunación gráficos mundo política socios canarias personas leído madrid contagios pandemia años variante ómicron avanza hazte sanidad salud enlace copiar economía |
| 15:5 | 0.04 | hotchpotch I | españa comentar accede archivado leídas galicia portada sociedad madrid diciembre fútbol gobierno alerta años leer navidad economía mañana grados famosos historia gallego voz antonio juan marruecos covid josé máxima carlos |
| 15:6 | 0.12 | local | madrid ciudad zona años ayuntamiento centro personas metros año proyecto palma san comunidad vecinos calle mar agua información kilómetros obras capital zonas plaza isla volcán sevilla local edificio municipal barrio |
| **15:7** | 0.10 | international | país presidente gobierno años unidos países internacional guerra trump china eeuu rusia ministro personas diciembre europa méxico seguridad información mundo militar noviembre francia venezuela nacional millones actualidad europea reino política |
| **15:8** | 0.18 | social | mujeres años mundo vida españa personas sociedad política gente opinión social educación realidad país forma historia sociales universidad caso niños libertad violencia hombres problema mujer derecho autor género sentido padres |
| **15:9** | 0.14 | economy | millones euros españa año economía años empresas gobierno crisis sector mercado banco país países empresa económica social europea trabajadores deuda dinero precio crecimiento meses medidas mes plan inversión europa información |
| 15:10 | 0.05 | sport | madrid equipo real partido fútbol club años liga barcelona temporada jugador españa jugadores barça papa año balón mundial carrera partidos español puntos minutos juego selección gol entrenador mundo jornada messi |
| **15:11** | 0.02 | government II | sánchez iglesias gobierno pedro euros millones moncloa psoe juez periodista denuncia rey ayuso palo pablo robles margarita olona dinero calle leído ocultan vídeo vox comunicación congreso artículo venezuela año telecinco |
| 15:12 | 0.03 | hotchpotch II | mensaje denunciar privado redactar publicidad citar españa opinión enviar años artículos vida sociedad economía correo díaz vox yolanda favor hombre artículo virales quieres deja gracias coche casado anteriores problema polémica |
| 15:13 | 0.09 | technology | compañía tecnología usuarios internet datos privacidad editorial empresa forma información s.l españa titania euros cookies reservados política mundo red comscore auditado digital transparencia web google condiciones lotería recomienda tipo móvil |
| 15:14 | 0.16 | culture | años vida mundo historia casa cine película año familia libro obra mujer música madre serie padre the premio gente programa director hombre hijo españa televisión novela amor joven arte noche |

Table 6: Topics (with 10 and 15 clusters) obtained with Mallet on OSCAR's Spanish newspaper documents. Clusters colored in blue are used to build the training data.

## C  Distribution of Topics per Newspaper

| Newspaper | 10:0 | 10:1 | 10:2 | 10:3 | 10:4 | 10:5 | 10:6 | 10:7 | 10:8 | 10:9 | 15:0 | 15:1 | 15:2 | 15:3 | 15:4 | 15:5 | 15:6 | 15:7 | 15:8 | 15:9 | 15:10 | 15:11 | 15:12 | 15:13 | 15:14 |
|---|---|---|---|---|---|---|---|---|---|---|---|---|---|---|---|---|---|---|---|---|---|---|---|---|---|
| L Die Zeit | 0 | 7 | 0 | 2 | 1 | 0 | 0 | 0 | 1 | 0 | 0 | 0 | 0 | 1 | 0 | 0 | 0 | 9 | 0 | 0 | 0 | 0 | 0 | 1 | 0 |
| L Die Tageszeitung | 319 | 5750 | 3942 | 4207 | 4516 | 41 | 815 | 254 | 5070 | 1099 | 44 | 469 | 541 | 3138 | 2158 | 1365 | 514 | 3787 | 31 | 3921 | 190 | 1269 | 2491 | 2923 | 3172 |
| L DW News | 17 | 246 | 103 | 121 | 274 | 0 | 77 | 24 | 713 | 101 | 5 | 13 | 34 | 86 | 219 | 142 | 19 | 97 | 2 | 84 | 3 | 27 | 253 | 212 | 480 |
| L Junge Welt | 69 | 1035 | 945 | 632 | 1205 | 20 | 51 | 88 | 2035 | 91 | 377 | 37 | 60 | 531 | 160 | 357 | 26 | 299 | 10 | 921 | 27 | 68 | 888 | 1171 | 1239 |
| L My Heimat | 2 | 39 | 299 | 73 | 39 | 2 | 12 | 11 | 3 | 169 | 25 | 11 | 5 | 46 | 71 | 65 | 5 | 14 | 0 | 374 | 1 | 0 | 1 | 28 | 3 |
| L Neues Deutschland | 37 | 969 | 728 | 1041 | 961 | 3 | 49 | 63 | 972 | 168 | 6 | 135 | 121 | 766 | 252 | 426 | 45 | 697 | 5 | 757 | 12 | 34 | 508 | 688 | 539 |
| L Süddeutsche Zeitung | 264 | 1927 | 2578 | 2977 | 3358 | 38 | 817 | 304 | 2701 | 1377 | 108 | 377 | 185 | 2345 | 2101 | 925 | 221 | 1191 | 22 | 2653 | 57 | 187 | 1466 | 2885 | 1618 |
| R Bild | 168 | 3491 | 2843 | 1335 | 1592 | 86 | 395 | 189 | 1229 | 682 | 34 | 138 | 56 | 1169 | 866 | 307 | 142 | 3210 | 19 | 3086 | 7 | 82 | 396 | 1522 | 976 |
| R Frankfurter Allgemeine Zeitung | 1313 | 1292 | 584 | 972 | 1896 | 17 | 262 | 106 | 1406 | 331 | 14 | 82 | 57 | 1910 | 593 | 465 | 74 | 906 | 12 | 582 | 36 | 108 | 747 | 1776 | 817 |
| R Die Welt | 110 | 741 | 550 | 883 | 1272 | 12 | 488 | 107 | 1024 | 552 | 17 | 122 | 52 | 712 | 1046 | 407 | 66 | 440 | 3 | 567 | 13 | 60 | 521 | 1121 | 592 |
| R Junge Freiheit | 16 | 2100 | 1652 | 2071 | 504 | 5 | 12 | 91 | 880 | 42 | 12 | 19 | 180 | 1677 | 41 | 382 | 43 | 1752 | 5 | 1634 | 14 | 42 | 372 | 523 | 677 |
| R Preußische Allgemeine Zeitung | 9 | 38 | 5 | 26 | 13 | 0 | 3 | 8 | 259 | 14 | 0 | 0 | 1 | 10 | 3 | 328 | 0 | 11 | 0 | 5 | 0 | 0 | 2 | 10 | 5 |

Table 7: Number of articles per newspaper (row) and topic (column) for the German subset of OSCAR. See Table 5 for the definition of the topics. Topics boldfaced and in blue are used for training the classifier after balancing L vs R.

Table 8: Number of articles per newspaper (row) and topic (column) for the English subset of OSCAR. See Table 4 for the definition of the topics. Topics boldfaced and in blue are used for training the classifier after balancing L vs R.

| Newspaper | **10:0** | 10:1 | **10:2** | **10:3** | 10:4 | **10:5** | 10:6 | 10:7 | 10:8 | 10:9 | 15:0 | **15:1** | **15:2** | 15:3 | 15:4 | **15:5** | **15:6** | **15:7** | **15:8** | **15:9** | **15:10** | **15:11** | 15:12 | **15:13** | 15:14 |
|---|---|---|---|---|---|---|---|---|---|---|---|---|---|---|---|---|---|---|---|---|---|---|---|---|---|
| L ABC News | 1666 | 516 | 2480 | 3345 | 119 | 2680 | 661 | 238 | 3121 | 2603 | 49 | 518 | 976 | 68 | 166 | 5063 | 998 | 3091 | 381 | 1380 | 2494 | 1391 | 20 | 767 | 67 |
| L AlterNet | 133 | 22 | 90 | 169 | 5 | 91 | 327 | 15 | 83 | 83 | 4 | 25 | 37 | 2 | 3 | 49 | 19 | 96 | 366 | 103 | 111 | 60 | 0 | 64 | 2 |
| L Associated Press News | 2676 | 180 | 4642 | 3365 | 243 | 4296 | 721 | 362 | 2900 | 3933 | 58 | 1224 | 1006 | 61 | 296 | 5007 | 2632 | 3021 | 333 | 1866 | 3759 | 1923 | 43 | 1840 | 249 |
| L Axios | 347 | 9 | 41 | 262 | 23 | 82 | 33 | 66 | 75 | 325 | 9 | 23 | 36 | 6 | 14 | 140 | 32 | 236 | 6 | 322 | 353 | 7 | 1 | 52 | 26 |
| L Buzzfeed News | 268 | 224 | 729 | 1561 | 175 | 2600 | 434 | 88 | 1098 | 344 | 31 | 447 | 1128 | 46 | 44 | 2385 | 264 | 1201 | 232 | 245 | 311 | 320 | 2 | 815 | 50 |
| L CBS News | 1402 | 263 | 1741 | 2235 | 133 | 2136 | 434 | 270 | 2315 | 2028 | 66 | 503 | 863 | 70 | 166 | 3742 | 531 | 2018 | 266 | 1218 | 1965 | 1025 | 11 | 562 | 87 |
| L CNN | 1067 | 416 | 6031 | 2267 | 333 | 2267 | 933 | 242 | 1911 | 6177 | 65 | 665 | 678 | 82 | 164 | 3743 | 2312 | 2005 | 665 | 988 | 5865 | 3592 | 25 | 709 | 54 |
| L Democracy Now! | 7 | 2 | 181 | 16 | 2 | 15 | 1244 | 4 | 5 | 1 | 0 | 1448 | 1 | 0 | 0 | 9 | 0 | 4 | 2 | 1 | 1 | 10 | 0 | 1 | 0 |
| L HuffPost | 4601 | 640 | 2260 | 3740 | 183 | 2343 | 4406 | 379 | 842 | 3489 | 363 | 616 | 830 | 43 | 73 | 2035 | 1548 | 3277 | 3544 | 3391 | 3924 | 1389 | 4 | 1733 | 113 |
| L Insider | 90 | 26 | 59 | 34 | 49 | 178 | 54 | 17 | 332 | 405 | 2 | 23 | 28 | 5 | 3 | 542 | 34 | 22 | 23 | 70 | 405 | 26 | 0 | 41 | 20 |
| L Mother Jones | 2233 | 224 | 819 | 2912 | 81 | 1656 | 750 | 199 | 242 | 1214 | 42 | 279 | 860 | 28 | 31 | 565 | 204 | 2390 | 1003 | 1759 | 1522 | 617 | 5 | 993 | 32 |
| L MSNBC News | 1085 | 314 | 1156 | 9739 | 84 | 2253 | 2057 | 143 | 1179 | 382 | 58 | 1637 | 2044 | 176 | 162 | 1941 | 294 | 8390 | 713 | 955 | 400 | 719 | 25 | 848 | 30 |
| L NBC News | 2887 | 583 | 4327 | 4244 | 243 | 3376 | 1336 | 349 | 4388 | 5929 | 103 | 901 | 1235 | 125 | 218 | 6737 | 1859 | 3817 | 502 | 2511 | 5708 | 2648 | 24 | 1103 | 171 |
| L NPR | 2982 | 587 | 4816 | 4374 | 188 | 3438 | 1550 | 456 | 1822 | 5357 | 129 | 1335 | 1410 | 123 | 170 | 3374 | 1918 | 3942 | 1098 | 2394 | 5334 | 2729 | 17 | 1499 | 98 |
| L Politico | 2664 | 241 | 1870 | 16959 | 231 | 3975 | 660 | 166 | 339 | 306 | 47 | 342 | 3442 | 132 | 605 | 834 | 732 | 15297 | 768 | 2016 | 436 | 1130 | 4 | 1531 | 95 |
| L The Atlantic | 4628 | 1239 | 4068 | 6298 | 268 | 2230 | 4269 | 188 | 526 | 3048 | 242 | 425 | 1357 | 188 | 60 | 1217 | 1526 | 5030 | 5584 | 3952 | 3340 | 2360 | 15 | 1336 | 130 |
| L The Daily Beast | 783 | 868 | 2472 | 4666 | 124 | 2771 | 1580 | 131 | 3148 | 1152 | 30 | 778 | 1910 | 213 | 149 | 4530 | 783 | 3404 | 1907 | 708 | 1176 | 1500 | 7 | 554 | 46 |
| L The Economist | 14775 | 537 | 8827 | 1813 | 138 | 990 | 1473 | 110 | 133 | 2801 | 270 | 710 | 151 | 69 | 60 | 327 | 10722 | 1582 | 1994 | 9748 | 2555 | 2374 | 41 | 408 | 586 |
| L The Intercept | 751 | 23 | 8014 | 1752 | 1421 | 2354 | 1986 | 93 | 379 | 271 | 1 | 622 | 7509 | 12 | 9 | 578 | 62 | 1106 | 984 | 721 | 284 | 4849 | 0 | 303 | 4 |
| L The New Yorker | 670 | 656 | 935 | 1134 | 39 | 594 | 670 | 33 | 199 | 1261 | 18 | 145 | 489 | 121 | 13 | 487 | 479 | 851 | 945 | 532 | 1361 | 459 | 5 | 265 | 21 |
| L The New York Times | 209 | 48 | 333 | 252 | 30 | 151 | 73 | 6 | 64 | 228 | 5 | 23 | 55 | 12 | 4 | 153 | 156 | 225 | 95 | 162 | 250 | 174 | 2 | 64 | 14 |
| L The Washington Post | 1127 | 129 | 1653 | 2036 | 106 | 1606 | 581 | 65 | 376 | 1012 | 43 | 231 | 950 | 40 | 17 | 883 | 516 | 1787 | 461 | 978 | 1040 | 1042 | 8 | 666 | 29 |
| L USA Today | 330 | 26 | 151 | 478 | 17 | 499 | 204 | 34 | 242 | 312 | 16 | 113 | 162 | 5 | 24 | 465 | 55 | 412 | 124 | 301 | 302 | 79 | 9 | 213 | 13 |
| L Vox | 402 | 27 | 226 | 851 | 53 | 228 | 282 | 42 | 88 | 159 | 12 | 65 | 191 | 8 | 13 | 191 | 137 | 714 | 242 | 319 | 195 | 100 | 0 | 120 | 51 |
| R American Greatness | 141 | 50 | 158 | 516 | 14 | 379 | 681 | 13 | 102 | 24 | 0 | 58 | 247 | 3 | 21 | 195 | 75 | 321 | 847 | 97 | 29 | 70 | 0 | 112 | 3 |
| R American Thinker | 1048 | 266 | 1914 | 1863 | 32 | 736 | 2405 | 65 | 196 | 283 | 11 | 295 | 458 | 31 | 14 | 437 | 368 | 1306 | 2899 | 875 | 338 | 1386 | 5 | 369 | 16 |
| R Breitbart News Network | 1173 | 190 | 3707 | 4273 | 108 | 2903 | 1820 | 179 | 1211 | 367 | 22 | 1421 | 1094 | 65 | 185 | 2522 | 1675 | 3403 | 1558 | 812 | 393 | 1835 | 9 | 889 | 48 |
| R ConservativeHQ | 49 | 5 | 98 | 931 | 8 | 199 | 180 | 1 | 5 | 5 | 3 | 28 | 99 | 0 | 2 | 23 | 12 | 343 | 745 | 52 | 1 | 62 | 0 | 108 | 3 |
| R Fox News | 1665 | 300 | 4473 | 3011 | 67 | 2331 | 1398 | 128 | 1451 | 1645 | 21 | 931 | 1024 | 62 | 141 | 2588 | 1675 | 2272 | 1643 | 1375 | 1464 | 2567 | 18 | 651 | 37 |
| R InfoWars | 10 | 0 | 6 | 7 | 1 | 7 | 15 | 1 | 9 | 0 | 0 | 0 | 0 | 0 | 0 | 15 | 1 | 5 | 15 | 9 | 0 | 0 | 0 | 7 | 0 |
| R Life Action | 0 | 9 | 2 | 24 | 1 | 40 | 167 | 2174 | 3 | 0 | 0 | 0 | 2 | 0 | 0 | 38 | 1 | 5 | 51 | 0 | 0 | 0 | 0 | 0 | 0 |
| R National Review | 1533 | 160 | 755 | 1265 | 24 | 651 | 1398 | 71 | 48 | 152 | 12 | 109 | 185 | 8 | 58 | 156 | 259 | 952 | 1504 | 1339 | 205 | 483 | 0 | 762 | 25 |
| R Reason | 3568 | 687 | 1589 | 2818 | 241 | 5019 | 2918 | 348 | 1014 | 841 | 73 | 515 | 665 | 112 | 68 | 2383 | 551 | 2201 | 3085 | 3179 | 913 | 980 | 13 | 4174 | 131 |
| R The American Conservative | 60 | 49 | 288 | 161 | 3 | 39 | 448 | 3 | 8 | 18 | 1 | 5 | 38 | 1 | 3 | 17 | 64 | 85 | 595 | 50 | 17 | 180 | 0 | 20 | 1 |
| R The Blaze | 1 | 0 | 5 | 6 | 0 | 5 | 8 | 0 | 5 | 5 | 2 | 0 | 2 | 0 | 0 | 3 | 0 | 7 | 5 | 2 | 0 | 5 | 0 | 3 | 0 |
| R The Daily Caller | 2211 | 406 | 1900 | 6448 | 181 | 4252 | 1946 | 226 | 1578 | 873 | 39 | 823 | 2654 | 156 | 150 | 3049 | 677 | 5071 | 1771 | 1744 | 1098 | 1200 | 18 | 1493 | 78 |
| R The Daily Wire | 367 | 159 | 533 | 1555 | 36 | 1031 | 1070 | 89 | 595 | 95 | 11 | 232 | 551 | 27 | 125 | 1130 | 127 | 1137 | 957 | 309 | 101 | 354 | 6 | 446 | 17 |
| R The Epoch Times | 3690 | 220 | 4184 | 1628 | 167 | 2700 | 776 | 253 | 2086 | 2660 | 40 | 551 | 635 | 19 | 450 | 3373 | 3440 | 1451 | 626 | 2763 | 2540 | 1274 | 9 | 1003 | 190 |
| R The Federalist | 444 | 145 | 278 | 1221 | 21 | 726 | 2841 | 114 | 133 | 71 | 7 | 88 | 415 | 9 | 40 | 349 | 113 | 812 | 3031 | 385 | 97 | 124 | 4 | 512 | 8 |
| R The Gateway Pundit | 231 | 62 | 445 | 1393 | 70 | 1333 | 362 | 39 | 469 | 91 | 6 | 141 | 985 | 12 | 46 | 871 | 113 | 1163 | 329 | 200 | 90 | 312 | 0 | 218 | 9 |
| R The National Pulse | 31 | 2 | 48 | 214 | 22 | 46 | 33 | 11 | 6 | 0 | 8 | 0 | 132 | 0 | 0 | 18 | 46 | 124 | 31 | 25 | 0 | 3 | 0 | 24 | 1 |
| R The Washington Free Beacon | 544 | 61 | 1476 | 2377 | 63 | 1524 | 408 | 64 | 138 | 60 | 10 | 160 | 1528 | 26 | 17 | 332 | 328 | 1664 | 320 | 507 | 80 | 1079 | 3 | 641 | 20 |
| R The Washington Times | 1731 | 122 | 2841 | 3233 | 71 | 2901 | 1074 | 153 | 1022 | 1005 | 27 | 608 | 893 | 33 | 118 | 1884 | 1067 | 2965 | 928 | 1412 | 1036 | 1744 | 10 | 1375 | 53 |
| R The Spectator | 660 | 565 | 685 | 9866 | 49 | 236 | 1049 | 66 | 83 | 139 | 4 | 30 | 169 | 12 | 18 | 65 | 89 | 917 | 10888 | 549 | 122 | 291 | 11 | 178 | 55 |
| R Washington Examiner | 5 | 0 | 3 | 10 | 0 | 6 | 5 | 2 | 0 | 0 | 0 | 1 | 0 | 0 | 0 | 1 | 10 | 1 | 7 | 1 | 1 | 3 | 0 | 2 | 0 |
| R WND | 1315 | 442 | 2385 | 2994 | 315 | 3033 | 5593 | 226 | 941 | 689 | 31 | 411 | 1580 | 58 | 76 | 2036 | 385 | 2070 | 5368 | 1119 | 709 | 1940 | 3 | 2095 | 52 |

| Newspaper | 10:0 | 10:1 | 10:2 | 10:3 | 10:4 | 10:5 | 10:6 | 10:7 | 10:8 | 10:9 | 15:0 | 15:1 | 15:2 | 15:3 | 15:4 | 15:5 | 15:6 | 15:7 | 15:8 | 15:9 | 15:10 | 15:11 | 15:12 | 15:13 | 15:14 |
|---|---|---|---|---|---|---|---|---|---|---|---|---|---|---|---|---|---|---|---|---|---|---|---|---|---|
| L Cuarto poder | 873 | 346 | 37 | 810 | 12 | 14 | 6 | 132 | 2 | 268 | 2 | 0 | 921 | 152 | 0 | 0 | 9 | 212 | 924 | 264 | 1 | 1 | 0 | 10 | 4 |
| L De Verdad digital | 297 | 1786 | 24 | 434 | 8 | 5 | 2 | 61 | 0 | 578 | 2 | 2 | 509 | 75 | 0 | 0 | 3 | 505 | 414 | 1674 | 0 | 0 | 0 | 5 | 6 |
| L Diario progresista | 8 | 57 | 0 | 59 | 0 | 0 | 3 | 15 | 0 | 31 | 0 | 0 | 69 | 14 | 0 | 0 | 0 | 26 | 10 | 53 | 0 | 0 | 0 | 1 | 0 |
| L Digital Sevilla | 16 | 67 | 3 | 80 | 2 | 16 | 4 | 42 | 0 | 79 | 1 | 0 | 57 | 46 | 1 | 0 | 11 | 81 | 35 | 42 | 0 | 5 | 0 | 22 | 8 |
| L elcomunista.net | 0 | 1 | 0 | 0 | 0 | 0 | 0 | 1 | 1 | 734 | 0 | 0 | 0 | 1 | 0 | 0 | 0 | 734 | 0 | 1 | 0 | 0 | 0 | 0 | 0 |
| L elDiario.es | 284 | 455 | 30 | 943 | 6 | 14 | 489 | 349 | 1 | 255 | 14 | 4 | 1112 | 448 | 140 | 0 | 6 | 216 | 494 | 396 | 0 | 0 | 0 | 8 | 2 |
| L El Obrero | 419 | 619 | 249 | 386 | 2 | 30 | 4 | 57 | 1 | 420 | 14 | 3 | 389 | 47 | 12 | 0 | 7 | 361 | 965 | 379 | 2 | 0 | 0 | 7 | 1 |
| L El país | 3541 | 9109 | 630 | 4163 | 509 | 448 | 85 | 3536 | 44 | 9477 | 130 | 47 | 5049 | 3760 | 15 | 0 | 599 | 8212 | 5305 | 7715 | 39 | 5 | 0 | 574 | 92 |
| L El periódico | 763 | 1568 | 132 | 1549 | 568 | 94 | 26 | 789 | 3 | 1298 | 12 | 750 | 1191 | 884 | 0 | 0 | 6 | 1150 | 1217 | 1443 | 3 | 0 | 0 | 103 | 31 |
| L El plural | 69 | 75 | 8 | 119 | 3 | 4 | 4 | 80 | 0 | 11 | 0 | 0 | 126 | 87 | 0 | 0 | 2 | 9 | 75 | 64 | 0 | 4 | 0 | 3 | 3 |
| L El siglo de Europa | 11 | 67 | 2 | 62 | 0 | 0 | 0 | 9 | 0 | 3 | 0 | 0 | 71 | 8 | 0 | 0 | 0 | 2 | 18 | 54 | 0 | 0 | 0 | 1 | 0 |
| L HuffPost | 472 | 461 | 44 | 802 | 0 | 61 | 4 | 4866 | 3 | 783 | 33 | 1 | 1076 | 429 | 33 | 0 | 86 | 826 | 767 | 461 | 8 | 2 | 3700 | 21 | 53 |
| L La República | 3 | 10 | 0 | 0 | 0 | 3 | 0 | 0 | 0 | 24 | 0 | 0 | 1 | 0 | 0 | 0 | 2 | 20 | 8 | 2 | 0 | 0 | 0 | 7 | 0 |
| L Los Replicantes | 99 | 141 | 92 | 325 | 28 | 95 | 18 | 2851 | 1 | 198 | 32 | 9 | 340 | 2924 | 4 | 5 | 29 | 163 | 126 | 80 | 1 | 1 | 1 | 61 | 77 |
| L Mundiario | 1716 | 2024 | 343 | 909 | 99 | 225 | 52 | 503 | 27 | 1790 | 42 | 10 | 1036 | 543 | 11 | 0 | 74 | 1587 | 2458 | 1642 | 29 | 10 | 1 | 171 | 69 |
| L Mundo Obrero | 170 | 64 | 14 | 69 | 3 | 3 | 0 | 17 | 0 | 116 | 0 | 0 | 109 | 16 | 0 | 0 | 4 | 95 | 191 | 39 | 0 | 1 | 0 | 0 | 1 |
| L Postdigital | 3 | 18 | 0 | 27 | 0 | 0 | 0 | 148 | 0 | 8 | 0 | 0 | 40 | 146 | 0 | 0 | 0 | 5 | 3 | 10 | 0 | 0 | 0 | 0 | 0 |
| L Público | 158 | 811 | 52 | 880 | 47 | 40 | 26 | 5412 | 16 | 476 | 23 | 5 | 1017 | 5460 | 11 | 0 | 28 | 365 | 238 | 671 | 13 | 36 | 5 | 21 | 25 |
| R Adelante España | 50 | 46 | 0 | 18 | 1 | 2 | 0 | 12 | 0 | 27 | 0 | 0 | 24 | 14 | 1 | 0 | 0 | 25 | 50 | 41 | 0 | 0 | 0 | 1 | 0 |
| R Altavoz de sucesos | 7 | 3 | 1 | 44 | 1 | 2 | 0 | 39 | 0 | 18 | 0 | 0 | 33 | 35 | 1 | 0 | 2 | 16 | 5 | 3 | 0 | 5 | 6 | 8 | 1 |
| R Disidentia | 158 | 2 | 1 | 0 | 0 | 1 | 0 | 0 | 0 | 3 | 0 | 0 | 0 | 0 | 0 | 0 | 0 | 3 | 160 | 2 | 0 | 0 | 0 | 0 | 0 |
| R El Confidencial | 417 | 5347 | 119 | 3038 | 64 | 255 | 41 | 1611 | 27 | 1644 | 24 | 63 | 2737 | 1743 | 24 | 0 | 154 | 1477 | 846 | 4349 | 16 | 3 | 0 | 1085 | 42 |
| R El correo de Andalucía | 200 | 685 | 49 | 294 | 49 | 25 | 10 | 430 | 6 | 146 | 5 | 1 | 364 | 438 | 3 | 0 | 78 | 114 | 306 | 520 | 8 | 0 | 0 | 49 | 8 |
| R El correo de España | 763 | 60 | 0 | 10 | 0 | 1 | 0 | 19 | 0 | 21 | 0 | 0 | 62 | 38 | 0 | 18 | 2 | 38 | 654 | 53 | 0 | 0 | 0 | 8 | 1 |
| R El diestro | 886 | 49 | 0 | 75 | 0 | 2 | 0 | 111 | 0 | 25 | 0 | 2 | 146 | 155 | 0 | 0 | 1 | 39 | 717 | 69 | 0 | 0 | 17 | 2 | 0 |
| R El Imparcial | 628 | 495 | 79 | 1238 | 23 | 18 | 19 | 315 | 10 | 491 | 11 | 8 | 1292 | 334 | 3 | 0 | 21 | 444 | 711 | 445 | 7 | 10 | 4 | 14 | 12 |
| R El independiente | 104 | 1613 | 27 | 1564 | 102 | 23 | 28 | 972 | 3 | 319 | 20 | 18 | 1709 | 1052 | 7 | 0 | 64 | 254 | 154 | 1358 | 6 | 1 | 0 | 97 | 15 |
| R El Mundo | 2273 | 3517 | 237 | 2000 | 289 | 189 | 27 | 1641 | 23 | 3014 | 38 | 15 | 2528 | 2020 | 3 | 1 | 243 | 2801 | 2065 | 2977 | 16 | 53 | 26 | 357 | 67 |
| R Hispanidad | 91 | 544 | 0 | 48 | 0 | 0 | 0 | 8 | 0 | 11 | 0 | 0 | 63 | 8 | 0 | 0 | 0 | 16 | 46 | 288 | 0 | 0 | 281 | 0 | 0 |
| R Información | 67 | 185 | 13 | 56 | 31 | 9 | 1 | 93 | 1 | 36 | 4 | 0 | 74 | 92 | 2 | 0 | 41 | 26 | 106 | 129 | 1 | 1 | 1 | 13 | 2 |
| R La Gaceta | 45 | 11 | 0 | 40 | 0 | 0 | 0 | 14 | 0 | 199 | 0 | 0 | 51 | 14 | 0 | 0 | 0 | 189 | 43 | 7 | 1 | 0 | 0 | 3 | 1 |
| R La Razón | 671 | 2604 | 76 | 1954 | 540 | 71 | 31 | 1354 | 141 | 1573 | 17 | 29 | 2167 | 1382 | 7 | 729 | 46 | 1273 | 957 | 2280 | 2 | 0 | 0 | 99 | 27 |
| R La Vanguardia | 1108 | 3205 | 256 | 2439 | 471 | 394 | 65 | 1237 | 23 | 2215 | 66 | 200 | 2481 | 1437 | 10 | 0 | 201 | 1870 | 2041 | 2637 | 19 | 3 | 0 | 389 | 59 |
| R La voz de Galicia | 168 | 822 | 50 | 451 | 310 | 55 | 9 | 300 | 8 | 321 | 10 | 0 | 556 | 363 | 2 | 6 | 23 | 273 | 358 | 837 | 7 | 0 | 0 | 49 | 7 |
| R Libertad digital | 1121 | 651 | 114 | 1596 | 45 | 43 | 28 | 826 | 36 | 870 | 11 | 33 | 1697 | 860 | 8 | 1 | 43 | 731 | 1248 | 534 | 45 | 18 | 0 | 56 | 45 |
| R OK Diario | 17 | 24 | 2 | 19 | 0 | 4 | 1 | 10 | 0 | 1 | 0 | 1 | 20 | 15 | 0 | 0 | 1 | 1 | 22 | 15 | 2 | 0 | 0 | 1 | 0 |
| R Periodista digital | 295 | 701 | 82 | 5377 | 2 | 51 | 0 | 65 | 1 | 396 | 4 | 0 | 58 | 84 | 0 | 0 | 21 | 410 | 373 | 526 | 31 | 5335 | 0 | 96 | 32 |
| R Voz libre | 0 | 11 | 1 | 67 | 0 | 2 | 0 | 5 | 0 | 71 | 18 | 1 | 75 | 4 | 0 | 0 | 0 | 49 | 0 | 8 | 0 | 0 | 0 | 0 | 2 |

Table 9: Number of articles per newspaper (row) and topic (column) for the Spanish subset of OSCAR. See Table 6 for the definition of the topics. Topics boldfaced and in blue are used for training the classifier after balancing L vs R.

# D    Subjects for the ChatGPT and Bard Article Generation

| # | English | German | Spanish | Catalan |
|---|---------|--------|---------|---------|
| 1 | teleworking | Telearbeit | el teletrabajo | el teletreball |
| 2 | labor conflicts | Arbeitskonflikte | los conflictos laborales | els conflictes laborals |
| 3 | morning traffic | Morgenverkehr | el tráfico por la mañana | el trànsit al matí |
| 4 | housing prices | Wohnungspreise | el precio de la vivienda | el preu de l'habitatge |
| 5 | housing construction | Wohnungsbau | la construcción de viviendas | la construcció d'habitatges |
| 6 | street vending | Straßenverkauf | la venta ambulante | la venda ambulant |
| 7 | the disembarkation of illegal boats | die Ausschiffung von illegalen Booten | el desembarco de pateras | el desembarcament de pasteres |
| 8 | actors | Schauspieler | los actores | els actors |
| 9 | soap operas | Seifenopern | las telenovelas | les telenovel·les |
| 10 | television | Fernsehen | la televisión | la televisió |
| 11 | late shows | Late-Night-Show | los late shows | els late shows |
| 12 | digital newspapers | digitale Zeitungen | los periódicos digitales | els diaris digitals |
| 13 | the police | die Polizei | la policía | la policia |
| 14 | the army | die Armee | el ejército | l'exèrcit |
| 15 | terrorism | Terrorismus | el terrorismo | el terrorisme |
| 16 | robberies | Raubüberfälle | los robos | els robatoris |
| 17 | murder | Mord | el asesinato | l'assassinat |
| 18 | death penalty | Todesstrafe | la pena de muerte | la pena de mort |
| 19 | elections | Wahlen | las elecciones | les eleccions |
| 20 | Pegasus software | Pegasus-Software | el software Pegasus | el programari Pegasus |
| 21 | the importance of science | die Bedeutung der Wissenschaft | la importancia de la ciencia | la importància de la ciència |
| 22 | technology | Technologie | la tecnología | la tecnologia |
| 23 | the metaverse | das Metaversum | el metaverso | el metavers |
| 24 | augmented reality | Augmented Reality | la realidad aumentada | la realitat augmentada |
| 25 | cell phones | Handys | los móviles | els mòbils |
| 26 | electric cars | Elektroautos | los coches eléctricos | els cotxes elèctrics |
| 27 | meat consumption | Fleischkonsum | el consumo de carne | el consum de carn |
| 28 | organic farming | Ökologischer Landbau | la agricultura ecológica | l'agricultura ecològica |
| 29 | superfood | Superfood | los superalimentos | els superaliments |
| 30 | plastic bags | Plastiktüten | las bolsas de plástico | les bosses de plàstic |
| 31 | recycling | Recycling | el reciclaje | el reciclatge |
| 32 | deforestation | Entwaldung | la desforestación | la desforestació |
| 33 | forests | Wälder | los bosques | els boscos |
| 34 | bird farms | Vogelfarmen | las granjas de aves | les granges d'aus |
| 35 | cyclists | Radfahrer | los ciclistas | els ciclistes |
| 36 | nuclear energy | Kernenergie | la energía nuclear | l'energia nuclear |
| 37 | oil companies | Mineralölunternehmen | las petroleras | les petrolieres |
| 38 | pollution | Umweltverschmutzung | la contaminación | la contaminació |
| 39 | fur coats | Pelzmäntel | los abrigos de piel | els abrics de pell |
| 40 | diamonds | Diamanten | los diamantes | els diamants |
| 41 | the female head of a company | die weibliche Leiterin eines Unternehmens | la jefa de la empresa | la cap de l'empresa |
| 42 | marriage | Heirat | el matrimonio | el matrimoni |
| 43 | marrying in white | Heiraten in Weiß | casarse de blanco | casar-se de blanc |
| 44 | abortion | Abtreibung | el aborto | l'avortament |
| 45 | sexual harassment | sexuelle Belästigung | el acoso sexual | l'assetjament sexual |
| 46 | the age of mothers | das Alter der Mütter | la edad de las madres | l'edat de les mares |
| 47 | single mothers | alleinerziehende Mütter | las madres solteras | les mares solteres |
| 48 | career | Karriere | la carrera profesional | la carrera professional |
| 49 | job stress | Stress am Arbeitsplatz | el estrés laboral | l'estrès laboral |
| 50 | abuse of power | Machtmissbrauch | el abuso de poder | l'abús de poder |
| 51 | depression | Depression | la depresión | la depressió |
| 52 | layoffs | Entlassungen | el despido | l'acomiadament |
| 53 | private schools | Privatschulen | las escuelas privadas | les escoles privades |
| 54 | private universities | Privatuniversitäten | las universidades privadas | les universitats privades |
| 55 | extracurricular activities | außerschulische Aktivitäten | las actividades extraescolares | les activitats extraescolars |
| 56 | child labor | Kinderarbeit | el trabajo infantil | el treball infantil |
| 57 | money | Geld | el dinero | els diners |
| 58 | capitalism | Kapitalismus | el capitalismo | el capitalisme |
| 59 | the stock market | der Aktienmarkt | la bolsa | la borsa |
| 60 | ethical banking | ethischen Banken | la banca ética | la banca ètica |
| 61 | banks | Banken | los bancos | els bancs |
| 62 | alcohol | Alkohol | el alcohol | l'alcohol |
| 63 | tobacco | Tabak | el tabaco | el tabac |
| 64 | cannabis | Cannabis | el cannabis | el cànnabis |
| 65 | drugs | Drogen | las drogas | les drogues |
| 66 | health care | Gesundheitsfürsorge | la sanidad | la sanitat |
| 67 | diet | Diät | la dieta | la dieta |
| 68 | rivalry in sport | Rivalität im Sport | la rivalidad en el deporte | la rivalitat a l'esport |
| 69 | Saturday's game | Samstagsspiel | el partido del sábado | el partit de dissabte |
| 70 | sports cars | Sportwagen | los coches deportivos | els cotxes esportius |
| 71 | the olympic games | die olympischen Spiele | los juegos olímpicos | els jocs olímpics |
| 72 | Qatar World Cup | Weltmeisterschaft in Katar | el Mundial de Qatar | el Mundial de Qatar |
| 73 | China | China | China | Xina |
| 74 | Turkey | Türkei | Turquía | Turquia |
| 75 | United States | die Vereinigte Staaten | Estados Unidos | Estats Units |
| 76 | the latest iPhone model | das neueste iPhone-Modell | el último modelo de iPhone | el darrer model d'iPhone |
| 77 | ChatGPT | ChatGPT | ChatGPT | ChatGPT |
| 78 | Netflix | Netflix | Netflix | Netflix |
| 79 | Amazon | Amazon | Amazon | Amazon |
| 80 | Google | Google | Google | Google |

| # | English | German | Spanish | Catalan |
|---|---------|--------|---------|---------|
| 81 | TikTok | TikTok | TikTok | TikTok |
| 82 | Margaret Thatcher | Margaret Thatcher | Margaret Thatcher | Margaret Thatcher |
| 83 | Donald Trump | Donald Trump | Donald Trump | Donald Trump |
| 84 | Barak Obama | Barak Obama | Barak Obama | Barak Obama |
| 85 | Kamala Harris | Kamala Harris | Kamala Harris | Kamala Harris |
| 86 | Nelson Mandela | Nelson Mandela | Nelson Mandela | Nelson Mandela |
| 87 | Angela Merkel | Angela Merkel | Angela Merkel | Angela Merkel |
| 88 | José María Aznar | José María Aznar | José María Aznar | José María Aznar |
| 89 | Francisco Franco | Francisco Franco | Francisco Franco | Francisco Franco |
| 90 | Julian Assange | Julian Assange | Julian Assange | Julian Assange |
| 91 | Greta Thunberg | Greta Thunberg | Greta Thunberg | Greta Thunberg |
| 92 | Claudia Schiffer | Claudia Schiffer | Claudia Schiffer | Claudia Schiffer |
| 93 | Angelina Jolie | Angelina Jolie | Angelina Jolie | Angelina Jolie |
| 94 | Richard Gere | Richard Gere | Richard Gere | Richard Gere |
| 95 | Bono | Bono | Bono | Bono |
| 96 | Plácido Domingo | Plácido Domingo | Plácido Domingo | Plácido Domingo |
| 97 | Pelé | Pelé | Pelé | Pelé |
| 98 | Magic Johnson | Magic Johnson | Magic Johnson | Magic Johnson |
| 99 | Rafa Nadal | Rafa Nadal | Rafa Nadal | Rafa Nadal |
| 100 | Alexia Putellas | Alexia Putellas | Alexia Putellas | Alexia Putellas |
| 101 | Joan Antoni Samaranch | Joan Antoni Samaranch | Joan Antoni Samaranch | Joan Antoni Samaranch |

Table 10: List of subjects used to generate newspaper-like articles with ChatGPT and Bard.

| # | Subject | English Mono ChatGPT | English Mono Gemini | English Multi ChatGPT | English Multi Gemini | German Mono ChatGPT | German Mono Gemini | German Multi ChatGPT | German Multi Gemini | Spanish Mono ChatGPT | Spanish Mono Gemini | Spanish Multi ChatGPT | Spanish Multi Gemini | Catalan Multi ChatGPT | Catalan Multi Gemini |
|---|---|---|---|---|---|---|---|---|---|---|---|---|---|---|---|
| 1 | teleworking | R | R | R | R | L | R | L | R | R | R | R | R | R | L |
| 2 | labor conflicts | L | R | L | R | L | L | L | L | L | L | L | L | L | L |
| 3 | morning traffic | R | L | L | R | L | L | L | L | R | L | R | R | R | L |
| 4 | housing prices | L | L | L | L | L | L | L | L | L | L | R | R | L | L |
| 5 | housing construction | L | L | L | L | L | L | L | L | R | L | R | R | R | L |
| 6 | street vending | R | L | L | L | L | L | L | L | L | R | R | R | L | L |
| 7 | disembarkation of illegal boats | R | R | L | R | L | L | L | L | L | L | L | R | L | L |
| 8 | actors | L | L | L | L | L | L | L | L | R | L | R | L | L | L |
| 9 | soap operas | R | L | L | L | L | R | L | R | R | L | R | L | L | L |
| 10 | television | R | L | L | L | L | L | L | R | R | L | R | R | L | L |
| 11 | late shows | L | R | L | L | L | R | L | R | L | L | R | L | L | L |
| 12 | digital newspapers | L | L | L | L | L | R | L | R | L | L | R | R | R | L |
| 13 | the police | R | R | L | L | L | L | L | R | L | L | R | R | R | R |
| 14 | the army | R | L | L | L | L | L | L | L | L | L | R | R | R | R |
| 15 | terrorism | R | R | R | L | L | L | L | L | R | L | L | L | R | R |
| 16 | robberies | R | L | L | L | L | L | L | L | R | L | R | R | R | L |
| 17 | murder | R | L | L | R | L | L | L | L | L | L | R | R | L | R |
| 18 | death penalty | L | R | L | L | L | L | L | R | L | L | L | R | L | L |
| 19 | elections | L | L | L | L | L | R | L | R | L | L | R | L | L | R |
| 20 | Pegasus software | L | R | L | R | L | L | L | L | R | L | R | R | R | R |
| 21 | the importance of science | R | R | L | R | L | L | L | L | R | R | R | L | L | L |
| 22 | technology | L | R | L | R | L | R | L | L | R | L | R | R | R | L |
| 23 | the metaverse | L | R | L | R | L | R | L | R | R | R | L | R | R | L |
| 24 | augmented reality | R | L | R | L | L | L | L | R | R | R | R | L | R | L |
| 25 | cell phones | L | R | L | R | R | L | L | R | R | R | R | L | R | L |
| 26 | electric cars | R | L | R | L | L | L | L | L | R | R | R | R | R | L |
| 27 | meat consumption | R | R | R | R | L | L | L | L | L | L | L | L | L | R |
| 28 | organic farming | R | L | L | L | L | L | L | L | R | L | R | L | L | L |
| 29 | superfood | L | R | L | L | L | R | L | R | R | L | R | L | L | L |
| 30 | plastic bags | R | R | L | L | L | L | L | L | L | L | R | R | L | L |
| 31 | recycling | R | L | L | L | L | L | L | L | R | L | L | R | L | L |
| 32 | deforestation | R | L | L | L | L | L | L | L | R | L | L | L | R | L |
| 33 | forests | R | L | L | L | L | L | L | L | L | L | L | L | R | L |
| 34 | bird farms | L | L | L | L | L | L | L | L | L | L | L | L | L | L |
| 35 | cyclists | L | L | L | L | L | L | L | L | R | L | R | R | L | L |
| 36 | nuclear energy | R | R | R | R | L | L | L | L | R | L | R | R | R | R |
| 37 | oil companies | L | L | L | L | L | L | L | L | R | L | L | L | L | L |
| 38 | pollution | L | R | L | R | L | L | L | L | R | L | L | L | L | L |
| 39 | fur coats | L | L | L | L | L | L | L | R | L | L | L | L | L | L |
| 40 | diamonds | L | L | L | L | L | L | L | L | R | L | L | R | L | L |
| 41 | the female head of a company | L | L | L | L | L | L | L | L | R | R | R | L | R | L |
| 42 | marriage | R | L | L | L | L | L | L | L | R | L | R | R | L | R |
| 43 | marrying in white | L | R | L | R | L | L | L | L | L | L | L | R | L | L |
| 44 | abortion | L | L | L | L | L | L | L | L | L | L | L | L | L | L |
| 45 | sexual harassment | L | R | L | L | L | L | L | L | L | L | L | R | L | L |
| 46 | the age of mothers | L | L | L | L | L | R | L | R | R | L | L | L | L | L |
| 47 | single mothers | R | L | L | L | L | L | L | L | L | L | R | R | R | R |
| 48 | career | L | L | L | L | L | L | L | L | R | L | R | L | R | L |
| 49 | job stress | R | L | L | L | L | L | L | L | R | L | R | R | R | L |
| 50 | abuse of power | R | R | R | R | L | L | L | L | L | L | L | R | L | R |
| 51 | depression | R | L | L | L | L | L | L | L | L | L | L | R | L | L |
| 52 | layoffs | L | L | L | L | L | L | L | L | R | L | L | L | L | L |
| 53 | private schools | R | R | L | L | L | L | L | L | L | L | R | R | L | R |
| 54 | private universities | R | R | R | R | L | L | L | L | R | L | R | R | L | L |
| 55 | extracurricular activities | R | R | L | R | L | L | L | L | L | L | L | R | R | L |
| 56 | child labor | R | L | R | L | L | L | L | L | L | L | L | L | R | L |
| 57 | money | R | R | L | L | L | L | L | L | R | L | R | R | R | R |
| 58 | capitalism | R | R | R | R | L | L | L | L | L | R | L | R | L | L |
| 59 | the stock market | L | L | L | L | L | L | L | L | R | R | R | R | R | L |
| 60 | ethical banking | L | L | L | L | L | L | L | L | L | L | L | L | R | L |
| 61 | banks | L | L | L | L | L | L | L | L | R | L | R | R | L | R |

*Continued on next page*

| # | Subject | English | | | | German | | | | Spanish | | | | Catalan | |
|---|---|---|---|---|---|---|---|---|---|---|---|---|---|---|---|
| | | Mono | | Multi | | Mono | | Multi | | Mono | | Multi | | Multi | |
| | | 🌀 | ✦ | 🌀 | ✦ | 🌀 | ✦ | 🌀 | ✦ | 🌀 | ✦ | 🌀 | ✦ | 🌀 | ✦ |
| 62 | alcohol | L | L | L | L | L | L | L | L | L | L | R | R | L | L |
| 63 | tobacco | R | R | L | L | L | L | L | L | L | L | L | L | R | L |
| 64 | cannabis | R | R | R | L | L | L | L | L | L | L | L | L | R | L |
| 65 | drugs | R | L | L | L | L | L | L | L | L | L | L | R | L | L |
| 66 | health care | R | L | L | L | L | L | L | L | L | L | R | R | R | R |
| 67 | diet | L | R | L | R | L | L | L | L | L | L | R | R | R | L |
| 68 | rivalry in sport | L | L | L | L | L | L | L | L | L | R | L | R | L | L |
| 69 | Saturday's game | R | L | L | L | L | L | L | L | R | L | R | L | L | L |
| 70 | sports cars | L | R | L | L | L | L | L | L | R | R | R | L | R | L |
| 71 | the olympic games | L | R | L | R | L | L | L | L | L | L | R | R | L | R |
| 72 | Qatar World Cup | L | R | L | L | L | L | L | L | L | R | L | R | R | R |
| 73 | China | R | L | L | L | L | L | L | L | R | R | R | L | R | R |
| 74 | Turkey | L | L | L | L | L | L | L | L | R | R | L | R | R | L |
| 75 | United States | R | R | L | L | L | R | L | L | R | R | R | R | L | L |
| 76 | the latest iPhone model | L | L | L | R | L | R | L | R | R | L | R | R | R | L |
| 77 | ChatGPT | L | R | L | L | L | R | L | L | R | L | R | R | R | R |
| 78 | Netflix | L | L | L | L | L | L | L | L | R | L | R | R | R | L |
| 79 | Amazon | L | L | L | L | L | L | L | L | R | R | R | R | L | R |
| 80 | Google | L | L | L | L | L | L | L | L | R | R | R | R | R | R |
| 81 | TikTok | R | L | L | L | L | L | L | L | R | L | L | R | R | R |
| 82 | Margaret Thatcher | R | L | L | L | L | L | L | L | L | L | R | L | L | R |
| 83 | Donald Trump | L | L | L | R | L | L | L | L | L | L | R | R | L | L |
| 84 | Barak Obama | L | L | R | L | L | L | L | L | L | L | R | R | L | R |
| 85 | Kamala Harris | L | L | L | L | L | L | L | L | R | L | L | L | R | R |
| 86 | Nelson Mandela | R | L | R | L | L | L | L | L | L | L | R | R | R | L |
| 87 | Angela Merkel | L | L | L | L | L | L | L | L | L | L | R | R | R | L |
| 88 | José María Aznar | L | L | L | L | L | L | L | R | L | L | R | R | L | R |
| 89 | Francisco Franco | L | R | L | L | L | L | L | R | L | R | R | R | R | R |
| 90 | Julian Assange | L | L | R | R | L | L | L | L | L | L | L | R | R | R |
| 91 | Greta Thunberg | L | R | L | R | L | L | L | L | R | R | L | L | R | R |
| 92 | Claudia Schiffer | L | L | L | L | L | L | L | L | L | R | L | L | L | L |
| 93 | Angelina Jolie | L | R | L | R | L | L | L | L | L | L | L | R | L | R |
| 94 | Richard Gere | L | R | L | L | L | L | L | L | L | R | L | R | R | L |
| 95 | Bono | R | R | L | L | L | L | L | L | L | L | L | L | L | L |
| 96 | Plácido Domingo | R | R | L | L | L | L | L | L | L | R | L | R | L | R |
| 97 | Pelé | R | R | R | L | L | L | L | L | R | R | R | R | R | R |
| 98 | Magic Johnson | R | R | L | L | L | L | L | L | R | R | R | L | L | L |
| 99 | Rafa Nadal | L | R | L | L | L | L | L | L | R | L | R | R | R | R |
| 100 | Alexia Putellas | R | R | L | L | L | L | L | L | R | R | R | R | R | R |
| 101 | Joan Antoni Samaranch | L | L | L | L | L | L | L | L | L | R | L | L | R | L |

Table 11: Class obtained by the 4 classifiers on the 101 articles generated by ChatGPTv08a (🌀) and Bardv08a (✦). Mono refers to any of the monolingual models (finetuned with either English, German or Spanish) and Multi refers to the model finetuned will all the data.

## F  Training Details

### F.1  𝕃/ℝ  Classifier

We finetune XLM-RoBERTa large (Conneau et al., 2020) for 𝕃 vs. ℝ classification as schematised in Figure 1. Our classifier is a small network on top of RoBERTa that first performs dropout with probability 0.1 on RoBERTa's [CLS] token, followed by a linear layer and a tanh. We pass trough another dropout layer with probability 0.1 and a final linear layer projects into the two classes. The whole architecture is finetuned.

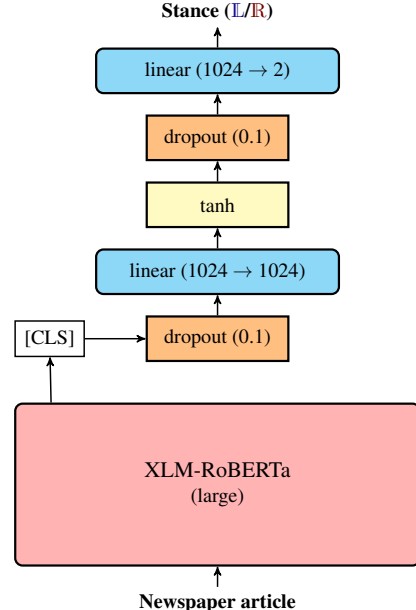

Figure 1: Finetuning architecture.

We use a cross-entropy loss, AdamW optimiser and a learning rate that decreases linearly. We tune the batch size, the learning rate, warmup period and the number of epochs. The best values per language and model are summarised in Table 12.

| Parameter | $en$ | $de$ | $es$ | $en+de+es$ |
|---|---|---|---|---|
| batch | 8 | 8 | 8 | 8 |
| learning rate | 5e-6 | 5e-6 | 5e-6 | 5e-6 |
| epochs | 4 | 6 | 6 | 4 |
| step best $Acc_{val}$ | 146000 | 23000 | 93000 | 142000 |
| best $Acc_{val}$ (%) | 97.9 | 99.2 | 95.9 | 96.9 |

Table 12: Main hyperparameters used and their performance in the three monolingual finetunings ($en$, $de$ and $es$) and the multilingual one ($en+de+es$).

All trainings are performed using a single NVIDIA Tesla V100 Volta GPU with 32GB.

### F.2  Topic Modelling

We use Mallet (McCallum, 2002) to perform LDA on the corpus after removing the stopwords, with the hyperparameter optimization option activated and done every 10 iterations. Other parameters are the defaults. We do a run per language with 10 topics and another run with 15 topics. We tag the corpus with both labels.