# OpenReview forum: "Multilingual Coarse Political Stance Classification of Media. The Editorial Line of a ChatGPT and Bard Newspaper"
_EMNLP/2023/Conference — EMNLP 2023 Findings_

### Official Review · Reviewer_3VUk · 2023-08-06

**Soundness:** 4

**Excitement:**

3: Ambivalent: It has merits (e.g., it reports state-of-the-art results, the idea is nice), but there are key weaknesses (e.g., it describes incremental work), and it can significantly benefit from another round of revision. However, I won't object to accepting it if my co-reviewers champion it.

**Paper Topic And Main Contributions:**

The authors present the results of a series of political stance classification experiments with the goal of identifying political bias in ChatGPT.  A corpus of multilingual newspaper articles tagged using distant supervision and then filtered for politically divisive articles (abortion, the death penalty, etc.) is used to train a binary political stance classifier, which achieves 95% accuracy on a held-out test set. The classifier is then passed over a series of ChatGPT-authored articles on politically divisive issues. The results show a left political bias across all languages in the test data.

**Reasons To Accept:**

- Diagnosing bias in LLMs, particularly ChatGPT, is a timely research topic and the authors have made an interesting attempt at providing a different approach to quantifying the bias in ChatGPT that has been reported extensively in the social science literature.

- The paper is overall well-written and sensibly organized

**Reasons To Reject:**

- The authors have reproduced a well-known result in the literature--left political bias in ChatGPT and in LLMs in general--using the "coarse" (their description) methodology of passing a binary stance classifier over ChatGPT's output. The observation that language models reproduce the biases of the corpora on which they're trained has been made at each step of the evolution of these models, from word2vec to BERT to ChatGPT, and so it's unclear why this observation needs to once again be made using the authors "coarse" methodology.

- The authors' decision to filter by divisive topic using introduces an unacknowledged prior: for most of the topics given in the appendix (immigration, the death penalty, etc.), the choice to bother writing about the topic itself introduces bias. This choice will be reflected in both the frequency with which such articles appear and in the language used in those articles. The existence of the death penalty, for example, might be considered unproblematic on the right and, for that reason, one will see fewer articles on the subject in right-leaning newspapers.  When they do appear, neutral language will be employed. The opposite is likely true for a left-leaning publication, for whom the existence of the death penalty is a problem: the articles will occur with more frequency and will employ less neutral language. For these reasons, it's not surprising that the authors' classifier, which was annotated via distant supervision using political slant tags assigned by Wikipedia, will tend to score most content generated by an LLM as left-leaning.

**Reproducibility:**

3: Could reproduce the results with some difficulty. The settings of parameters are underspecified or subjectively determined; the training/evaluation data are not widely available.

**Reviewer Confidence:**

4: Quite sure. I tried to check the important points carefully. It's unlikely, though conceivable, that I missed something that should affect my ratings.

---

> ### Author Rebuttal · Authors · 2023-08-27
>
> Thanks for the comments. We would like to first clarify two points of the "Paper Topic And Main Contributions" that do not correspond to what we reflect in the paper:
>
> - "filtered for politically divisive articles (abortion, the death penalty, etc.)"
>
> Abortion, the death penalty, but also Olympic games, Richard Gere, etc. are the subjects we use to generate articles with chatGPT (Table 7), not for filtering the training data (Sec2-ChatGPT Corpus). For filtering, we apply LDA on the data and obtain broad topics that we label as "International", "Government", "Law & Justice", "Sports"... (Tables 4-6, and Sec2-Topic Modelling), then, we select a subset of the broad topics for training as explained in Section 2.
>
> - "The results show a left political bias across all languages in the test data."
>
> Our main conclusion already summarised in the abstract is "ChatGPT editorial line evolves with time and, being a data-driven system, the stance of the generated articles differs among languages." The analysis Section 3 with the results in Table 2 show that, for instance, the last chatGPT version we analyse (v05) has a right-leaning in English and Spanish. In the Conclusion we state "We show that ChatGPT writes newspaper articles with a Right-oriented perspective for English and Spanish, and results seem to indicate a Left orientation for German and Catalan."
>
> These misunderstandings are related to the reasons to reject the paper we would like to comment:
>
> **1a)** We are not reproducing previous results, as we don't claim that chatGPT is biased towards the left, but that the leaning depends on the language and the period. We are providing a classifier to easily check the stance of a new version when needed.
>
> **1b)** Still in the same paragraph, we agree that LMs reproduce the biases of the training data, but in the case of chatGPT we don't know the training data (and even less the human annotations for the reinforcement learning from human feedback phase), so, under our point of view, it makes sense to study the biases it has.
>
> **2a)** We agree that a filtering that only selects papers about immigration or death penalty would introduce a bias but, as explained above, this is not what we are doing: we test, not filter, on those topics (and others such as "technology", "television", "the Olympic games"...). In the whole, the training corpus is balanced in the number of articles left vs right.
>
> **2b)** "When they do appear, neutral language will be employed. The opposite is likely true for a left-leaning publication, for whom the existence of the death penalty is a problem: the articles will occur with more frequency and will employ less neutral language." Assuming that articles about death penalty appear in the training data, the model could hopefully learn as you say that if the language is not neutral it is a left-oriented article and if the language is neutral is right-oriented. And this is what we want. Now, if we use the classifier on chatGPT we will know if chatGPT's language is more or less neutral.
>
> **2c)** "which was annotated via distant supervision using political slant tags assigned by Wikipedia". This is not what we say in the paper, in Section 2 we write "We do not manually annotate any article, but we trust AllSides, MB/FC, Political Watch and Wikipedia (in cases where the information is not available in the previous sites) with their classification of a newspaper bias." Wikipedia is a backup for the sites that do bias annotation in a professional manner, and that we used in 4 sources and double checked with natives, as Germany is mostly not covered by those sites. We can add this information (where the leaning of every source comes from) in Table 3. We would like to stress that our main sources are AllSides, MB/FC and Political Watch.
>
> Finally, we would like to notify to the 3 reviewers that during the reviewing period 2 very interesting things happened: 1) a new version of chatGPT was deployed and 2) a multilingual Bard is now available. With this we can extend our results (Table 2) with 4 more rows: 2 generations with the newest version of chatGPT (named v08a and v08b in the table below) and 2 generations with Bard. Our aim is to reflect the variability (or lack of!) among generations with a same chatGPT/Bard version in a systematic manner (it was just a footnote in the submitted version as noticed by one reviewer) and, to the best of our knowledge, we are the first ones to analyse Bard's bias multilingualy.
>
> A very interesting thing to notice in the new results is that the period of time when chatGPT produced more right-oriented content corresponds to the time when chatGPT experienced a performance drop in several tasks according to [1]. The old and new results can be seen in the table below.
>
> |                   | English     |||| German       |||| Spanish     |||| Catalan  ||
> |-------------------|------------------------|-------------------------|------------------------|-------------------------|------------------------|-------------------------|-------------------------|----------|--------|---------|--------|---------|--------|---------|
> |                   | Mono || Multi || Mono || Multi ||Mono || Multi || Multi ||
> | (\% of articles)  | L                 | R                 | L                 | R                 | L                 | R                 | L                  | R  | L | R | L | R | L | R |
> | chatGPT v02       | **75&plusmn;9**         | 25&plusmn;8                | **93&plusmn;5**         | 7&plusmn;5                | --                     | --                      | --                      | -- | **65&plusmn;10**   | 35&plusmn;10              | **53&plusmn;10**         | 47&plusmn;10              | --                      | --                     |
> | chatGPT v03       | --                     | --                      | --                     | --                      | **97&plusmn;4**         | 3&plusmn;3                 | **69&plusmn;9**          | 31&plusmn;9 | --     | --      | --     | --      | --     | --      |
> | chatGPT v05       | 26&plusmn;9               | **74&plusmn;9**          | 40&plusmn;9               | **60&plusmn;9**          | **96&plusmn;5**         | 4&plusmn;3                 | **65&plusmn;9**          | 35&plusmn;9     | 25&plusmn;9               | **75&plusmn;9**          | 26&plusmn;9               | **74&plusmn;8**          | **71&plusmn;9**         | 29&plusmn;9                |
> | chatGPT v08a      | **54&plusmn;10**        | 46&plusmn;10               | **85&plusmn;8**         | 15&plusmn;6                | **99&plusmn;3**         | 1&plusmn;1                 | **100&plusmn;2**         | 0&plusmn;0                    | 50&plusmn;10              | 50&plusmn;10               | 40&plusmn;10              | **60&plusmn;10**         | 50&plusmn;10              | 50&plusmn;9                |
> | chatGPT v08b      | **52&plusmn;10**        | 48&plusmn;10               | **85&plusmn;8**         | 15&plusmn;6                | **100&plusmn;2**        | 0&plusmn;0                 | **100&plusmn;2**         | 0&plusmn;0                  | **51&plusmn;10**        | 49&plusmn;10               | 36&plusmn;10              | **64&plusmn;9**          | 47&plusmn;10              | **53&plusmn;10**         |
> | bard v08a         | **57&plusmn;11**        | 43&plusmn;10               | **75&plusmn;9**         | 25&plusmn;8                | **82&plusmn;8**         | 18&plusmn;7                | **82&plusmn;8**          | 18&plusmn;7                   | **74&plusmn;9**         | 26&plusmn;8                | 35&plusmn;9               | **65&plusmn;9**          | **66&plusmn;9**         | 34&plusmn;9                |
> | bard v08b         | **61&plusmn;10**        | 39&plusmn;10               | **82&plusmn;8**         | 18&plusmn;7                | **81&plusmn;8**         | 19&plusmn;7                | **90&plusmn;7**          | 10&plusmn;5 | **74&plusmn;9**         | 26&plusmn;8                | 44&plusmn;10              | **56&plusmn;10**         | **68&plusmn;9**         | 32&plusmn;9                |
>
>
>
> [1] Chen, Lingjiao, Matei Zaharia, and James Zou. "How is ChatGPT's behavior changing over time?." arXiv preprint arXiv:2307.09009 (2023).

---

### Official Review · Reviewer_75sN · 2023-08-06

**Typos Grammar Style And Presentation Improvements:** 1- formatting of the numbers, e.g. 74…
**Soundness:** 3

**Excitement:**

3: Ambivalent: It has merits (e.g., it reports state-of-the-art results, the idea is nice), but there are key weaknesses (e.g., it describes incremental work), and it can significantly benefit from another round of revision. However, I won't object to accepting it if my co-reviewers champion it.

**Paper Topic And Main Contributions:**

The authors measure political stance (left vs right) of ChatGPT over a couple of periods. A corpus of newspapers is collected based on its political bias (left vs right) in multiple languages and refined using LDA. The corpus is utilized to develop several classifiers for political stances. This classifier is used to determine stance of news articles created by ChatGPT in a couple of periods.

**Questions For The Authors:**

1- Instead of using LDA, simple filtering based on a category assigned by the source could be used (I know it is tricky for many sources, but you do not need many sources for the aims of this study). A simple classifier could be a straightforward tool for filtering as well.

2- Topic detection may be affected by the ideology of the source. What were the parameters of the topic modeling, how many times did it run, etc.?

3- How much of the data was discarded using LDA-based filtering?

4- Is the data open (copyright, etc.)? Or do you plan to release at least the URLs of the articles?

**Reasons To Accept:**

1- The research question is interesting.

2- The methodology chosen is a proper one.

3- The newspaper corpus is valuable. It is multilingual. The ChatGPT-generated news articles are valuable as well.

**Reasons To Reject:**

1- It is hard to understand the paper. There are many gaps in the report.

2- The evaluation is not clear.

**Reproducibility:**

2: Would be hard pressed to reproduce the results. The contribution depends on data that are simply not available outside the author's institution or consortium; not enough details are provided.

**Reviewer Confidence:**

3: Pretty sure, but there's a chance I missed something. Although I have a good feel for this area in general, I did not carefully check the paper's details, e.g., the math, experimental design, or novelty.

---

> ### Author Rebuttal · Authors · 2023-08-28
>
> First of all thanks for all your comments. Let us clarify some concepts and answer your questions below (we use QX for questions and PX for presentation improvements). We will extend the appendix with more statistics of the corpus to cover some of your comments. If there is the opportunity during the rebuttal, it would be very useful for us if you could specify the gaps in the report (reason to reject) or if those are related to the LDA questions which we address here. It would be also useful to know which are the dark points in the evaluation, so that we can clarify them (basically we use accuracy to evaluate our classifiers).
>
> Some of the comments and questions can be explained by the nature of the training data, which has been extracted from OSCAR [1] an open source project which does an additional cleaning to Common Crawl, a repository of web crawled data [2].
>
> **Q1**: We use OSCAR because it contains data in 166 languages and the data there is open; given that, anyone can easily extend the current analysis to more languages. We extracted 1,258,212 articles from 116 newspapers (Sec.2), but being an extensive web crawl, there is no information about the category of each article. We find important to have as many sources (newspapers) as possible in order to improve generalisation. Training a classifier to detect the category implies pre-specifying the categories and using some annotated data for training (which we don't have in general). Even if we think this is the way to go when dealing in depth with one language, the aim of our method is to go multilingual, cross-cultural and facilitate the analysis in any language. That includes languages for which no annotated data with categories is freely available. Also notice that with LDA, topics differ across languages (German has the topic we label as Nazism and Immigration appears in English but not in the others for example). We can capture this only with unsupervised methods, a newspaper would never categorise an article this way.
>
> **Q2**: Do you mean that the topics are different for the left and for the right newspapers? We studied this when creating the corpus to be sure that the corpus is balanced or at least that all classes are well represented. We will add the tables in the new appendix. To summarise: what happens is that the variation in newspapers across a same leaning is larger than the difference between L and R, although it is true that there exist differences. For instance, for the left in English we consider among others The Economist (mostly talking about "International" and "Economy") and Politico (mostly talking about "Government"), but when considering all the sources, all topics are covered. For this reason it is important to have as many sources as possible. Another example, in average, the largest difference of topic in Spanish newspapers is given for topic "Law and Justice" (15,024 articles L vs 10,068 R), so even if there is a difference, both leanings are well represented. Finally, let us point out that we expect the classifier not to learn only the content but also the style of L and R, we do the selection by topic only to minimise the number of neutral articles. When we generate the chatGPT outputs, we use more topics such as those related to Sport or Science which have not been seen during training but could have a different style depending on the leaning of the writer.
>
> We use Mallet to perform LDA on the corpus after removing the stopwords, with the hyperparameter optimization option activated and done every 10 iterations. Other parameters are the defaults. We do a run per language with 10 topics and another run with 15 topics. We tag the corpus with both labels.
>
>
> **Q3**: The percentage depends on the language. The method filters out 49% of the Spanish articles, 39% of the German ones and 31% of the English ones. We will add this information per source and topic in the new appendix.
>
> **Q4**: The data is a processed subset of OSCAR, so it will be freely available in the same terms as OSCAR. For every article, we will provide the OSCAR ID, the link to the article as given by OSCAR, the class we assign to the article according to its source (L/R), our cleaned text, and our 2 topic classes.
>
> **P3**: We manually inspected OSCAR's articles to detect patterns for newspaper-specific headers and footers. We eliminated these lines in the articles with regular expressions. The regular expressions and the patterns will be available in the Github together with the main code so that all the process can be reproduced.
>
> **P4**: We defined 101 subjects in English to prompt chatGPT (left column of Table 7). Then, we asked native speakers of German, Spanish and Catalan to manually translate them. This process is bias free, as only the concept itself (i.e., "technology") is translated.
>
> **P5**: We tested 3 versions of chatGPT (v02, v03 and v05 in Tables 1 and 2) but only v05 is simultaneously tested for the 4 languages. This changed now, as during the reviewing period a new version appeared which has been also tested for the 4 languages (see v08 below).
>
> **P6**: For German, the result was exactly the same (L=97&plusmn;4, R=3&plusmn;3 with both generations); for Catalan, the result lied within 95% confidence intervals (generation1: L=47&plusmn;10, R=53&plusmn;9 vs generation 2: L=50&plusmn;10, R=50&plusmn;9). We will remove this footnote and add the results for 2 generations in 4 languages with the new chatGPT version (see v08 below).
>
> **P7**: This is what Table 2 shows, but there is probably a misunderstanding between training and test which we will clarify. For training, a source (i.e., The Economist) is either L or R at 100%. For The Economist all the articles are labelled as Left articles in our training data. This is the assumption of our method, and the main reason why we try to filter out articles prone to be neutral in a newspaper. At testing time, we classify every article independently and look at the percentage in the test newspaper or chatGPT generations. This is Table 2. There cannot be an average across newspapers: we only use a newspaper for testing as the main goal is chatGPT, the newspaper we test is only to see whether the classifier is able to generalise to unseen newspapers or not as this has been shown to be a hard task [3]. We could average chatGPT versions, but one of our conclusions is precisely that it changes with time.
>
> Finally, we would like to notify to the 3 reviewers that during the reviewing period 2 very interesting things happened: 1) a new version of chatGPT was deployed as we said above and 2) a multilingual Bard is now available. With this we can extend our results (Table 2) with 4 more rows: 2 generations with the newest version of chatGPT (named v08a and v08b in the table below) and 2 generations with Bard. Our aim is to reflect the variability (or lack of!) among generations with a same chatGPT/Bard version in a systematic manner and substitute the footnote in the submitted paper (P6 improvement above) by the complete results. Also, and to the best of our knowledge, we will be the first ones to analyse Bard's bias multilingually. This experiment will enlarge our corpus of multilingual AI-generated newspaper articles that can be used for further research.
>
> A very interesting thing to notice in the new results is that the period of time when chatGPT produced more right-oriented content corresponds to the time when chatGPT experienced a performance drop in several tasks according to [4]. The old and new results can be seen in the table below.
>
> |                   | English     |||| German       |||| Spanish     |||| Catalan  ||
> |-------------------|------------------------|-------------------------|------------------------|-------------------------|------------------------|-------------------------|-------------------------|----------|--------|---------|--------|---------|--------|---------|
> |                   | Mono || Multi || Mono || Multi ||Mono || Multi || Multi ||
> | (\% of articles)  | L                 | R                 | L                 | R                 | L                 | R                 | L                  | R  | L | R | L | R | L | R |
> | chatGPT v02       | **75&plusmn;9**         | 25&plusmn;8                | **93&plusmn;5**         | 7&plusmn;5                | --                     | --                      | --                      | -- | **65&plusmn;10**   | 35&plusmn;10              | **53&plusmn;10**         | 47&plusmn;10              | --                      | --                     |
> | chatGPT v03       | --                     | --                      | --                     | --                      | **97&plusmn;4**         | 3&plusmn;3                 | **69&plusmn;9**          | 31&plusmn;9 | --     | --      | --     | --      | --     | --      |
> | chatGPT v05       | 26&plusmn;9               | **74&plusmn;9**          | 40&plusmn;9               | **60&plusmn;9**          | **96&plusmn;5**         | 4&plusmn;3                 | **65&plusmn;9**          | 35&plusmn;9     | 25&plusmn;9               | **75&plusmn;9**          | 26&plusmn;9               | **74&plusmn;8**          | **71&plusmn;9**         | 29&plusmn;9                |
> | chatGPT v08a      | **54&plusmn;10**        | 46&plusmn;10               | **85&plusmn;8**         | 15&plusmn;6                | **99&plusmn;3**         | 1&plusmn;1                 | **100&plusmn;2**         | 0&plusmn;0                    | 50&plusmn;10              | 50&plusmn;10               | 40&plusmn;10              | **60&plusmn;10**         | 50&plusmn;10              | 50&plusmn;9                |
> | chatGPT v08b      | **52&plusmn;10**        | 48&plusmn;10               | **85&plusmn;8**         | 15&plusmn;6                | **100&plusmn;2**        | 0&plusmn;0                 | **100&plusmn;2**         | 0&plusmn;0                  | **51&plusmn;10**        | 49&plusmn;10               | 36&plusmn;10              | **64&plusmn;9**          | 47&plusmn;10              | **53&plusmn;10**         |
> | bard v08a         | **57&plusmn;11**        | 43&plusmn;10               | **75&plusmn;9**         | 25&plusmn;8                | **82&plusmn;8**         | 18&plusmn;7                | **82&plusmn;8**          | 18&plusmn;7                   | **74&plusmn;9**         | 26&plusmn;8                | 35&plusmn;9               | **65&plusmn;9**          | **66&plusmn;9**         | 34&plusmn;9                |
> | bard v08b         | **61&plusmn;10**        | 39&plusmn;10               | **82&plusmn;8**         | 18&plusmn;7                | **81&plusmn;8**         | 19&plusmn;7                | **90&plusmn;7**          | 10&plusmn;5 | **74&plusmn;9**         | 26&plusmn;8                | 44&plusmn;10              | **56&plusmn;10**         | **68&plusmn;9**         | 32&plusmn;9                |
>
> [1] https://oscar-project.org/
>
> [2] https://commoncrawl.org/
>
> [3] [We Can Detect Your Bias: Predicting the Political Ideology of News Articles](https://aclanthology.org/2020.emnlp-main.404) (Baly et al., EMNLP 2020)
>
> [4] Chen, Lingjiao, Matei Zaharia, and James Zou. "How is ChatGPT's behavior changing over time?." arXiv preprint arXiv:2307.09009 (2023).

---

### Official Review · Reviewer_2C5A · 2023-08-10

**Soundness:** 4

**Excitement:**

4: Strong: This paper deepens the understanding of some phenomenon or lowers the barriers to an existing research direction.

**Paper Topic And Main Contributions:**

This paper analyses whether a hypothetical ChatGPT-authored news source would be considered left or right-leaning, if media observers assessed it. It covers English, German, Spanish and Catalan.

Its main contributions are:
- Showing it is possible to determine the L/R stance of unseen real media outlets with a classifier trained on real news articles labelled with the L/R stance of their source. This trialled both monolingual and multilingual models, and zero-shot performance.
- A dataset of ChatGPT-authored news articles on contentious topics in the four languages, generated using multiple versions of ChatGPT
- Analysis of the classifier on the dataset, showing that all considered versions of ChatGPT exhibit strong bias. It shows that the bias varies across languages, and that this bias has flipped between different versions.

**Questions For The Authors:**

A: The average word counts of the data suggests a significant amount of it exceeds the maximum token length of the classification model. Was any consideration given to this? Whilst in newspaper articles, the most significant information is typically at the beginning, there is no guarantee that ChatGPT can replicate this structure.

B: During preprocessing, were self-references to the newspapers removed, as these would also allow classifiers to learn sources? e.g. "[SOURCE] was unable to reach x for comment", "x, speaking exclusively to [SOURCE]" etc.

C: What motivated the choice of languages? Particularly, why was it decided to use Catalan as zero-shot when Spanish is included, was trialling a less-related language as zero-shot considered?

--

_Thank you for your answers to these questions which I am satisfied with_

**Reasons To Accept:**

This paper provides an interesting analysis of the political bias of ChatGPT's output. This has been a contentious topic, with notable public figures criticising OpenAI's approach to reducing bias and improving safety, claiming they are inserting political biases into the model. This work demonstrates that the news output of ChatGPT is notably politically biased, and this bias has (in English, for example) flipped between the oldest and newest versions.

Particular strengths:
- Considering which news topics are most likely to be politically divisive, both in general and for specific cultures
- Evaluating multiple versions of ChatGPT, with consideration that other factors (article length) are not responsible for polarity shift. _This is furthered by data provided in the author response, which includes subsequent versions of ChatGPT and results for Bard_
- I found the methodology simple and easy to follow

**Reasons To Reject:**

This paper only considers binary Left/Right political stance, a simplified view of a highly complex problem. The authors note that the prior work of García-Díaz et al. using the same label schema is on social media posts, but do not discuss the implications of using the same simple labels on long-form journalistic texts. _Given the rebuttal I accept that this simplification is necessary. I still think that the paper would benefit from acknowledging this simplification is likely to be more significant for these long-form texts_

Not enough information is provided to reproduce the heuristic cleaning in the preprocessing stage - the code in the supplementary information does not appear to include this. _The authors have stated they will add this to the code, so I believe this is no longer an issue_

The prompt used in ChatGPT is very simplistic. Since it is instructed to create articles on a topic, rather than an event as real news would be, I am concerned about the stylistic differences this will cause. Reading just the first few articles in English, it appears many of the generated articles resemble 'blogspam' more than news (although there are some articles where it has created a fictional event to write about). It is a possibility that the linguistic properties of the generated articles (which may also change over time as the model is updated) are more similar to one leaning of news or the other, which the classifier could be using instead of the actual content of the articles.

 _Following the subsequent author comment I am satisfied that this is in fact a property of both the real and ChatGPT-generated corpora, so is less likely to be problematic_

**Reproducibility:**

4: Could mostly reproduce the results, but there may be some variation because of sample variance or minor variations in their interpretation of the protocol or method.

**Reviewer Confidence:**

4: Quite sure. I tried to check the important points carefully. It's unlikely, though conceivable, that I missed something that should affect my ratings.

**Typos Grammar Style And Presentation Improvements:**

I found the explanation of article length (lines 264-268) confusing initially. I think it would be clearer to represent the comparison in length distribution as a graph. Also, it discusses token length when table 1 used word count - I think this should be consistent.

I found table 2 hard to understand initially. Perhaps it would be clearer if the text "Classification (% of articles)" ran across the top of the data, to make it clearer the L and R columns are the predicted labels.

---

> ### Author Rebuttal · Authors · 2023-08-28
>
> Thanks a lot for your detailed review. We will of course address the comments regarding presentation improvements, thank you. Below some comments and answers to the other issues (Reasons to Reject with prefix RX, Questions: QX).
>
> **R1**: We completely agree that considering the problem as a binary (L/R) classification is a simplification of the problem and we discussed the issue in the Limitations section. We think it is a limitation both for short texts (García-Díaz et al.) and long texts (us), so that reference is not intended to be a justification but just to show what has been done before. Our motivation for the simplification is to be able to do an analysis that goes beyond US English text. The annotation of data is not only expensive in time and resources, it also needs a specific training for the annotators. To give an example, a specialist site as Ad Fontes [1] has 60 analysts which receive a 30h initial training plus an additional training of 40h/year. Besides, for every article, there must be the same number of annotators with self-identified bias from the Left, Right and Centre to get reliable results according to the AllSides site [2]. Doing this is not feasible for most research groups and, as a consequence, we mostly have analyses with US data. In our case, we try to simplify the problem to be able to cover more countries and languages in almost an automatic way. We show that the results of our binary classification in 4 languages are reliable, and it is easy to extend to other languages with few language knowledge, but the same method would not work with a 5 degrees scale. We believe that being able to give a coarse indication, just L vs R, for many languages can be useful. In our work, we wouldn't have been able to analyse ChatGPT in Spanish and Catalan otherwise for example.
>
> **R2**: Heuristics for cleaning. We manually inspected the data to detect patterns mainly for newspaper-specific headers and footers. We eliminated these lines in the articles with regular expressions, but this is not part of the main code. We will add the regular expressions and the patterns in the github together with the main code so that all the process can be reproduced.
>
> **R3**: ChatGPT prompts. The stylistic output of chatGPT is in fact a very interesting topic. We tried different prompts with no significant differences in the style, length or content generated, so we went with the most simple one. It might be that chatGPT was/is generating blog-like text and this deserves a follow-up work where we analyse and compare the stylistic features of real blogs, news and chatGPT when asked to generate blogs and news. The goal of this short paper is only the political leaning. I would consider the style also a characteristic of the leaning though, so if chatGPT style resembles more to left or right-oriented ways to talk we think this is also a valid feature to classify the generated text. Unfortunately, we cannot show and prove the following with a reference, but for the sake of the argument, in an independent work we did some simple stylistic analysis of the data available in the Manifesto Project [3] (political parties’ election manifestos) and obtain that right-leaning texts are less complex, include less numbers, etc. confirming that the style of the text is also a characteristic of the leaning.
>
>
> **QA**: Yes, a significant amount of real newspaper articles are above the 512 limit, but we expect articles to have the most relevant information at the beginning. You are right and this might not be the case for chatGPT but the generated articles are in general shorter: only the last version of chatGPT for Spanish and English is slightly longer. To be sure that the length is not relevant and the training data we use has the relevant information at the beginning, we implemented a version of the classifier that averages the embeddings of the document at sentence level before classification so that all the document is considered (--split_documents option in the code). We did not find significant differences on the experiments we performed but training time was significantly slower so we decided not to include this in the paper.
>
> **QB**: Not explicitly, but the patterns for the heuristics included strings such as "The AP news staff was not involved in its creation" or "data provided by Refinitiv Lipper." Our current corpus has 1.2M articles in 4 languages so it might be the case that some self-references are still there. We plan to provide another version of the corpus covering more countries and with another round of cleaning but 100% removal of these expressions multilingually will not be possible as this would imply going though the million(s) of articles manually.
>
> **QC**: We chose the languages for which we could get help from native speakers in our lab. We used these native speakers to manually translate from English the 101 subjects we use, and to check the adequacy of the prompts we provide to chatGPT. We also asked them feedback about the newspapers we are using for testing the classifier. We acknowledge that with our models zero-shot for Catalan will perform better than zero-shot for Icelandic and we tried to make it clear in the paper by specifying that the good performance is on close languages.
>
> Finally, we would like to notify to the 3 reviewers that during the reviewing period 2 very interesting things happened: 1) a new version of chatGPT was deployed and 2) a multilingual Bard is now available. With this we can extend our results (Table 2) with 4 more rows: 2 generations with the newest version of chatGPT (named v08a and v08b in the table below) and 2 generations with Bard. Our aim is to reflect the variability (or lack of!) among generations with a same chatGPT/Bard version in a systematic manner (it was just a footnote in the submitted version as noticed by one reviewer) and, to the best of our knowledge, we are the first ones to analyse Bard's bias multilingualy.
>
> A very interesting thing to notice in the new results is that the period of time when chatGPT produced more right-oriented content corresponds to the time when chatGPT experienced a performance drop in several tasks according to [4]. The old and new results can be seen in the table below.
>
> |                   | English     |||| German       |||| Spanish     |||| Catalan  ||
> |-------------------|------------------------|-------------------------|------------------------|-------------------------|------------------------|-------------------------|-------------------------|----------|--------|---------|--------|---------|--------|---------|
> |                   | Mono || Multi || Mono || Multi ||Mono || Multi || Multi ||
> | (\% of articles)  | L                 | R                 | L                 | R                 | L                 | R                 | L                  | R  | L | R | L | R | L | R |
> | chatGPT v02       | **75&plusmn;9**         | 25&plusmn;8                | **93&plusmn;5**         | 7&plusmn;5                | --                     | --                      | --                      | -- | **65&plusmn;10**   | 35&plusmn;10              | **53&plusmn;10**         | 47&plusmn;10              | --                      | --                     |
> | chatGPT v03       | --                     | --                      | --                     | --                      | **97&plusmn;4**         | 3&plusmn;3                 | **69&plusmn;9**          | 31&plusmn;9 | --     | --      | --     | --      | --     | --      |
> | chatGPT v05       | 26&plusmn;9               | **74&plusmn;9**          | 40&plusmn;9               | **60&plusmn;9**          | **96&plusmn;5**         | 4&plusmn;3                 | **65&plusmn;9**          | 35&plusmn;9     | 25&plusmn;9               | **75&plusmn;9**          | 26&plusmn;9               | **74&plusmn;8**          | **71&plusmn;9**         | 29&plusmn;9                |
> | chatGPT v08a      | **54&plusmn;10**        | 46&plusmn;10               | **85&plusmn;8**         | 15&plusmn;6                | **99&plusmn;3**         | 1&plusmn;1                 | **100&plusmn;2**         | 0&plusmn;0                    | 50&plusmn;10              | 50&plusmn;10               | 40&plusmn;10              | **60&plusmn;10**         | 50&plusmn;10              | 50&plusmn;9                |
> | chatGPT v08b      | **52&plusmn;10**        | 48&plusmn;10               | **85&plusmn;8**         | 15&plusmn;6                | **100&plusmn;2**        | 0&plusmn;0                 | **100&plusmn;2**         | 0&plusmn;0                  | **51&plusmn;10**        | 49&plusmn;10               | 36&plusmn;10              | **64&plusmn;9**          | 47&plusmn;10              | **53&plusmn;10**         |
> | bard v08a         | **57&plusmn;11**        | 43&plusmn;10               | **75&plusmn;9**         | 25&plusmn;8                | **82&plusmn;8**         | 18&plusmn;7                | **82&plusmn;8**          | 18&plusmn;7                   | **74&plusmn;9**         | 26&plusmn;8                | 35&plusmn;9               | **65&plusmn;9**          | **66&plusmn;9**         | 34&plusmn;9                |
> | bard v08b         | **61&plusmn;10**        | 39&plusmn;10               | **82&plusmn;8**         | 18&plusmn;7                | **81&plusmn;8**         | 19&plusmn;7                | **90&plusmn;7**          | 10&plusmn;5 | **74&plusmn;9**         | 26&plusmn;8                | 44&plusmn;10              | **56&plusmn;10**         | **68&plusmn;9**         | 32&plusmn;9                |
>
>
> [1] https://adfontesmedia.com/
>
> [2] https://www.allsides.com/unbiased-balanced-news
>
> [3] https://manifesto-project.wzb.eu/information/documents/information
>
> [4] Chen, Lingjiao, Matei Zaharia, and James Zou. "How is ChatGPT's behavior changing over time?." arXiv preprint arXiv:2307.09009 (2023).

---

### Meta-Review · Area_Chair_5xuw · 2023-09-18

**Recommendation:** 4

**Metareview:**

Reviewers agree that this paper is sound and moderately exciting, adding to existing work on understanding of political bias in LLMs. Reviewers raised some concerns about clarity and particular methodological choices; authors provided an extensive set of rebuttals including updated empirical results with newer multilingual models and satisfied many reviewer questions and concerns.

Reviewers noted several issues that could be clarified in any final version of the paper, including: a) that binary left/right political stance is a simplification and potential limitation with long-form text, b) that the corpus includes non-news articles, c) some further explanation on the heuristics used for data cleanliness, and so on. We ask the authors to integrate these changes and clarifications following reviewer feedback.

---

### Decision · Program_Chairs · 2023-10-07

**Decision:**

Accept-Findings

**Comment:**

Reviewers agree that this paper is sound and moderately exciting, adding to existing work on understanding of political bias in LLMs. Reviewers raised some concerns about clarity and particular methodological choices; authors provided an extensive set of rebuttals including updated empirical results with newer multilingual models and satisfied many reviewer questions and concerns.

Reviewers noted several issues that could be clarified in any final version of the paper, including: a) that binary left/right political stance is a simplification and potential limitation with long-form text, b) that the corpus includes non-news articles, c) some further explanation on the heuristics used for data cleanliness, and so on. We ask the authors to integrate these changes and clarifications following reviewer feedback.